# Global Convergence of Deep Networks with One Wide Layer Followed by Pyramidal Topology

**Quynh Nguyen**[*]
MPI-MIS, Germany
quynh.nguyen@mis.mpg.de

**Marco Mondelli**
IST Austria
marco.mondelli@ist.ac.at

## Abstract

Recent works have shown that gradient descent can find a global minimum for over-parameterized neural networks where the widths of all the hidden layers scale polynomially with $N$ ($N$ being the number of training samples). In this paper, we prove that, for deep networks, a single layer of width $N$ following the input layer suffices to ensure a similar guarantee. In particular, all the remaining layers are allowed to have constant widths, and form a pyramidal topology. We show an application of our result to the widely used LeCun's initialization and obtain an over-parameterization requirement for the single wide layer of order $N^2$.

## 1 Introduction

Training a neural network is NP-Hard in the worst case [8], and the optimization problem is non-convex with many distinct local minima [4, 34, 45]. Yet, in practice neural networks with many more parameters than training samples can be successfully trained using gradient descent methods [47]. Understanding this phenomenon has recently attracted a lot of interest within the research community.

In [22], it is shown that, for the limiting case of infinitely wide neural networks, the convergence of the gradient flow trajectory can be studied via the so-called 'Neural Tangent Kernel' (NTK). Other recent works study the convergence properties of gradient descent, but they consider only one-hidden-layer networks [11, 14, 17, 25, 31, 38, 44], or require all the hidden layers to scale polynomially with the number of samples [2, 16, 48, 49]. In contrast, neural networks as used in practice are typically only wide at the first layer(s), after which the width starts to decrease toward the output layer [20, 35]. Motivated by this fact, we study how gradient descent performs under this pyramidal topology. Into this direction, it has been shown that the loss function of this class of networks is well-behaved, e.g. all sublevel sets are connected [26], or a weak form of Polyak-Lojasiewicz inequality is (locally) satisfied [29]. However, no algorithmic guarantees have been provided so far in the literature.

**Main contributions.** We show that a single wide layer followed by a pyramidal topology suffices to guarantee linear convergence of gradient descent to a global optimum. More specifically, in our main result (Theorem 3.2) we identify a set of sufficient conditions on the initialization and the network topology under which the global convergence of gradient descent is obtained. In Section 3.1, we show that these conditions are satisfied when the network has $N$ neurons in the first layer and a constant (i.e., independent of $N$) number of neurons in the remaining layers, $N$ being the number of training samples. Section 3.2 shows an application of our theorem to the popular LeCun's initialization [19, 21, 24], in which case the width of the first layer scales roughly as $N^2/\lambda_*^2$, where $\lambda_*$ is the smallest eigenvalue of the expected feature output at the first layer. Lastly, in Section 3.3 we show that $\lambda_*$ scales as a constant (i.e., independent of $N$) for sub-Gaussian training data.

**Comparison with related work.** Table 1 summarizes existing results and compares them with ours. The focus here is on regression problems. For classification problems (with logistic loss) we refer

---

[*]This work was done partly while the author was at TU Kaiserslautern

Table 1: Convergence guarantees for gradient descent. $N$ is the number of training samples; $L$ is the network depth; $\lambda_0$ is the smallest eigenvalue of the Gram matrix for a two-layer network [17, 31] ($\lambda_0$ scales as a constant under suitable assumptions on the training data); $\phi$ is the minimum $L_2$ distance between any pair of training data points; $\lambda_{\min}(K^{(L)})$ is the smallest eigenvalue of the Gram matrix defined recursively for an $L$-layer network [16] (the dependence of $\lambda_{\min}(K^{(L)})$ on $(N, L)$ remains unclear); $\lambda_*$ is defined in (13) and we show that it scales as a constant for sub-Gaussian training data on the sphere. Prior works assume the training samples have unit norm, i.e. $\|x_i\| = 1$ for $x_i \in \mathbb{R}^d$, whereas, for LeCun's initialization, we assume the data has norm $\sqrt{d}$.

| | Deep? | Multiple Outputs? | Activation | Layer Width | Parame-terization | Train All Layers? | #Wide Layers |
|---|---|---|---|---|---|---|---|
| [31] | No | No | Smooth | $\Omega\left(N^2\lambda_0^{-2}\right)$ | NTK | No | x |
| [2] | Yes | Yes | General | $\Omega\left(N^{24}L^{12}\phi^{-4}\right)$ | Standard | No | All |
| [49] | Yes | No | ReLU | $\Omega\left(N^8 L^{12}\phi^{-4}\right)$ | Standard | No | All |
| [16] | Yes | No | Smooth | $\Omega\left(\frac{N^4 2^{\mathcal{O}(L)}}{\lambda_{\min}^4(K^{(L)})}\right)$ | NTK | Yes | All |
| **Ours (general)** | **Yes** | **Yes** | **Smooth** | $N$ | **Standard** | **Yes** | **One** |
| **Ours (LeCun)** | **Yes** | **Yes** | **Smooth** | $\Omega\left(N^2 2^{\mathcal{O}(L)}\lambda_*^{-2}\right)$ | **Standard** | **Yes** | **One** |

to [12, 23, 30, 39]. Note that a direct comparison is not possible since the settings of these works are different. The novelty here is that we require only the first layer to be wide, while previous works require all the hidden layers to be wide. Thus, we are able to analyze a more realistic network topology – the pyramidal topology [20]. Furthermore, we identify a class of initializations such that the requirement on the width of the first layer is only $N$ neurons. This is, to the best of our knowledge, the first time that such a result is proved for gradient descent, although it was known that a width of $N$ neurons suffices for achieving a well-behaved loss surface, see [26, 27, 28, 29]. For LeCun's initialization, our over-parameterization requirement is of order $N^2$, which matches the best existing bounds for shallow nets [31, 38]. Let us highlight that we consider the standard parameterization, as opposed to the NTK parameterization [13, 22] (see e.g. [32] for a discussion on their performance).

**Proof techniques.** The work of [16, 17] analyzes the Gram matrices of the various layers and shows that they tend to be independent of the network parameters. In [2, 48, 49], the authors obtain a local semi-smoothness property of the loss function and a lower bound on the gradient of the last hidden layer. These results share the same network topology as the NTK analysis [22], in the sense that all the hidden layers need to be simultaneously very large. In [31], the authors analyze the Jacobian of a two-layer network, but this appears to be intractable for multilayer architectures.

Our paper shares with prior work [17, 22, 31] the intuition that over-parameterization, under the square loss, makes the trajectory of gradient descent remain bounded. We then exploit the structure of the pyramidal topology via a corresponding version of the Polyak-Lojasiewicz (PL) inequality [29], and the fact that the gradient of the loss is locally Lipschitz continuous. Using these two properties, we obtain the linear convergence of gradient descent by using the well-known recipe in non-convex optimization [33]: "*Lipschitz gradient + PL-inequality ⟹ Linear convergence*".

We highlight that our non-convex optimization perspective allows us to consider more general settings than existing NTK analyses. In fact, if the width of one of the layers is constant, then the NTK is not well defined [36]. On the contrary, our paper just requires the first layer to be overparameterized (i.e., all the other layers can have constant widths). To obtain the result for LeCun's initialization, we show that the smallest eigenvalue of the expected feature output at the first layer scales as a constant. This requires a bound on the smallest singular value of the Khatri-Rao powers of a random matrix with sub-Gaussian rows, which may be of independent interest.

## 2 Problem Setup

We consider an $L$-layer neural network with activation function $\sigma : \mathbb{R} \to \mathbb{R}$ and parameters $\theta = (W_l)_{l=1}^L$, where $W_l \in \mathbb{R}^{n_{l-1} \times n_l}$ is the weight matrix at layer $l$. Given $\theta_a = (W_l^a)_{l=1}^L$ and $\theta_b =$

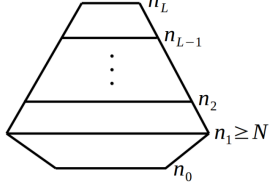

Figure 1: A network satisfying Assumption 2.1.

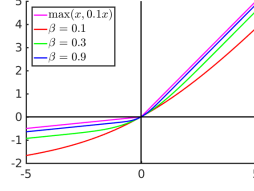

Figure 2: Activations (2) satisfying Assumption 2.2.

$(W_l^b)_{l=1}^L$, their $L_2$ distance is given by $\|\theta_a - \theta_b\|_2 = \sqrt{\sum_{l=1}^L \left\|W_l^a - W_l^b\right\|_F^2}$, where $\|\cdot\|_F$ denotes the Frobenius norm. Let $X \in \mathbb{R}^{N \times d}$ and $Y \in \mathbb{R}^{N \times n_L}$ be respectively the training input and output, where $N$ is the number of training samples, $d$ is the input dimension and $n_L$ is the output dimension (for consistency, we set $n_0 = d$). Let $F_l \in \mathbb{R}^{N \times n_l}$ be the output of layer $l$, which is defined as

$$F_l = \begin{cases} X & l = 0, \\ \sigma(F_{l-1}W_l) & l \in [L-1], \\ F_{L-1}W_L & l = L, \end{cases} \tag{1}$$

where $[L-1] = \{1, \dots, L-1\}$ and the activation function $\sigma$ is applied componentwise. Let $G_l = F_{l-1}W_l \in \mathbb{R}^{N \times n_l}$ for $l \in [L-1]$ and $G_L = F_L$ denote the pre-activation output. Let $f_l = \text{vec}(F_l) \in \mathbb{R}^{Nn_l}$ and $y = \text{vec}(Y) \in \mathbb{R}^{Nn_L}$ be obtained by concatenating their columns.

We are interested in minimizing the square loss $\Phi(\theta) = \frac{1}{2}\|f_L(\theta) - y\|_2^2$. To do so, we consider the gradient descent (GD) update $\theta_{k+1} = \theta_k - \eta \nabla \Phi(\theta_k)$, where $\eta$ is the step size and $\theta_k = (W_l^k)_{l=1}^L$ contains all parameters at step $k$.

In this paper, we consider a class of networks with one wide layer followed by a pyramidal topology, as studied in prior works [26, 28, 29] in the theory of optimization landscape (see also Figure 1).

**Assumption 2.1** *(Pyramidal network topology) Let $n_1 \geq N$ and $n_2 \geq n_3 \geq \dots \geq n_L$.*

Note that this assumption does not imply any ordering between $n_1$ and $n_2$. We make the following assumptions on the activation function $\sigma$.

**Assumption 2.2** *(Activation function) Fix $\gamma \in (0, 1)$ and $\beta > 0$. Let $\sigma$ satisfy that: (i) $\sigma'(x) \in [\gamma, 1]$, (ii) $|\sigma(x)| \leq |x|$ for every $x \in \mathbb{R}$, and (iii) $\sigma'$ is $\beta$-Lipschitz.*

As a concrete example, we consider the following family of parameterized ReLU functions, smoothened by a Gaussian kernel (see Figure 2 for an illustration):

$$\sigma(x) = -\frac{(1-\gamma)^2}{2\pi\beta} + \frac{\beta}{1-\gamma}\int_{-\infty}^{\infty}\max(\gamma u, u)\, e^{-\frac{\pi\beta^2(x-u)^2}{(1-\gamma)^2}}\, du. \tag{2}$$

The activation (2) satisfies Assumption 2.2 and it uniformly approximates the ReLU function over $\mathbb{R}$, i.e., $\lim_{\beta \to \infty} \sup_{x \in \mathbb{R}} |\sigma(x) - \max(\gamma x, x)| = 0$ (for the proof, see Lemma B.1 in Appendix B.1).

## 3 Main Results

First, let us introduce some notation for the singular values of the weight matrices at initialization:

$$\bar{\lambda}_l = \begin{cases} \frac{2}{3}(1 + \|W_l^0\|_2), & \text{for } l \in \{1, 2\}, \\ \|W_l^0\|_2, & \text{for } l \in \{3, \dots, L\}, \end{cases} \quad \lambda_l = \sigma_{\min}(W_l^0), \quad \lambda_{i \to j} = \prod_{l=i}^{j} \lambda_l, \quad \bar{\lambda}_{i \to j} = \prod_{l=i}^{j} \bar{\lambda}_l, \tag{3}$$

where $\sigma_{\min}(A)$ and $\|A\|_2$ are the smallest resp. largest singular value of the matrix $A$. We define $\lambda_F = \sigma_{\min}(\sigma(XW_1^0))$ as the smallest singular value of the output of the first hidden layer at initialization. We also make the following assumptions on the initialization.

**Assumption 3.1** *(Initial conditions)*

$$\lambda_F^2 \geq \frac{\gamma^4}{3}\left(\frac{6}{\gamma^2}\right)^L \|X\|_F \sqrt{2\Phi(\theta_0)}\frac{\bar{\lambda}_{3\to L}}{\lambda_{3\to L}^2}\max\left(\frac{2\bar{\lambda}_1\bar{\lambda}_2}{\min_{l\in\{3,\dots,L\}}\lambda_l\bar{\lambda}_l}, \bar{\lambda}_1, \bar{\lambda}_2\right), \tag{4}$$

$$\lambda_F^3 \geq \frac{2\gamma^4}{3}\left(\frac{6}{\gamma^2}\right)^L \|X\|_2 \|X\|_F \sqrt{2\Phi(\theta_0)}\frac{\bar{\lambda}_{3\to L}}{\lambda_{3\to L}^2}\bar{\lambda}_2. \tag{5}$$

Our main theorem is the following. Its proof is presented in Section 4.

**Theorem 3.2** *Let the network satisfy Assumption 2.1, the activation function satisfy Assumption 2.2 and the initial conditions satisfy Assumption 3.1. Define*

$$\alpha_0 = \frac{4}{\gamma^4}\left(\frac{\gamma^2}{4}\right)^L \lambda_F^2 \lambda_{3\to L}^2,$$

$$Q_0 = L\sqrt{L}\left(\frac{3}{2}\right)^{2(L-1)}\|X\|_F^2 \frac{\bar{\lambda}_{1\to L}^2}{\min_{l\in[L]}\lambda_l^2} + L\sqrt{L}\|X\|_F\left(1 + L\beta\|X\|_F R\right)R\sqrt{2\Phi(\theta_0)},\tag{6}$$

*with $R = \prod_{p=1}^L \max\left(1, \frac{3}{2}\bar{\lambda}_p\right)$. Let the learning rate be $\eta < \min\left(\frac{1}{\alpha_0}, \frac{1}{Q_0}\right)$. Then, the training loss vanishes at a geometric rate as*

$$\Phi(\theta_k) \leq (1 - \eta\alpha_0)^k \Phi(\theta_0). \tag{7}$$

*Furthermore, define*

$$Q_1 = \frac{4}{3}\left(\frac{3}{2}\right)^L \frac{\|X\|_F}{\alpha_0}\sum_{l=1}^L \frac{\bar{\lambda}_{1\to L}}{\bar{\lambda}_l}\sqrt{2\Phi(\theta_0)}. \tag{8}$$

*Then, the network parameters converge to a global minimizer $\theta_*$ at a geometric rate as*

$$\|\theta_k - \theta_*\|_2 \leq (1 - \eta\alpha_0)^{k/2}Q_1. \tag{9}$$

Theorem 3.2 shows that, for our pyramidal network topology, gradient descent converges to a global optimum under suitable initializations. Next, we discuss how these initial conditions may be satisfied.

## 3.1 Width $N$ suffices for a class of initializations

Let us show an example of an initialization that fulfills Assumption 3.1. Recall that, by Assumption 2.2, $\sigma(0) = 0$. Pick $W_1^0$ so that $\lambda_F$ is strictly positive[2]. One concrete example is $W_1^0$ chosen according to LeCun's initialization, i.e., $[W_1^0]_{ij} \sim \mathcal{N}(0, 1/d)$. Pick $(W_l^0)_{l=3}^L$ so that $\lambda_l \geq 1$ and $\lambda_l^2 \geq c\bar{\lambda}_l$ for every $l \in [3, L]$, for some $c > 1$. One concrete example is $[W_l^0]_{ij} \sim \mathcal{N}(0, (200c)^2/n_{l-1})$, which fulfils our requirements w.p. $\geq 1 - \sum_{l=3}^L e^{-\Omega(n_{l-1})}$ under the extra condition $\sqrt{n_{l-1}} \geq 1.01\sqrt{n_l}$. Another option is to pick $W_l^0$, for $3 \leq l \leq L$, to be scaled identity matrices (or rectangular matrices whose top block is a scaled identity). Next, set $W_2^0$ to be a random matrix with i.i.d. entries whose distribution has zero mean and variance $v$. Then, by choosing a sufficiently small $v$ (dependent on $c$), the following upper bounds hold with high probability:

$$\bar{\lambda}_2 = \frac{2}{3}(1 + \|W_2^0\|_2) \leq 1,$$

$$\sqrt{2\Phi(\theta_0)} \leq \|Y\|_F + \|F_L(\theta_0)\|_F \leq \|Y\|_F + \prod_{l=1}^L \|W_l^0\|_2 \|X\|_F \leq 2\|Y\|_F. \tag{10}$$

We note that a trivial choice of $v$ would be $v = 0$, which directly implies that (10) holds with probability 1. Now one observes that to satisfy (4), it suffices to have

$$\lambda_F^2\left(\frac{\gamma^4}{3}\left(\frac{6}{\gamma^2}\right)^L 2\|X\|_F \|Y\|_F \max\left(2\bar{\lambda}_1, 1\right)\right)^{-1} \geq \frac{\bar{\lambda}_{3\to L}}{\lambda_{3\to L}^2}. \tag{11}$$

The LHS of (11) depends only on $W_1^0$ and it is strictly positive as $\lambda_F > 0$, whereas the RHS of (11) depends only on $(W_l^0)_{l=3}^L$. Once the LHS stays fixed, the RHS can be made arbitrarily small by increasing the value of $c$ (and, consequently, decreasing the value of $v$). Thus, (11) is satisfied for $c$ large enough, and condition (4) holds. Similarly, we can show that condition (5) also holds. Note that this initialization does not introduce additional over-parameterization requirements. Hence, Theorem 3.2 requires only $N$ neurons at the first layer and allows a constant number of neurons in all the remaining layers.

In this example, the total number of parameters of the network is $\Omega(N)$, which is believed to be tight for memorizing $N$ *arbitrary* data points, see e.g. [5, 7, 18, 43, 46]. However, our result is not optimal in terms of layer widths. In fact, in [46] it is shown that, for a three-layer network, $\sqrt{N}$ neurons in each layer suffice for perfect memorization. Notice that [46] concerns the memorization capacity of neural networks, while we are interested in algorithmic guarantees.

As a technical remark, we note that if $x_i = x_j$ for $i \neq j$, then $\lambda_F = 0$. Thus, Assumption 3.2 cannot hold unless $\Phi = 0$ (i.e., we initialize at a global minimum) or $X = 0$ (i.e., the GD iterates do not move). In general, by using arguments along the lines of [16], one can show that, if the data points are not parallel and the activation function is analytic and not polynomial, then $\lambda_F > 0$. Furthermore, if $x_i$ and $x_j$ are close, then $\lambda_F$ is small and, therefore, $\alpha_0$ is small. Thus, GD requires more iterations to converge to a global optimum. This happens regardless of the value of $y_i$ and $y_j$. Providing results for deep pyramidal networks that depend on the quality of the labels is an outstanding problem. Solving it could also lead to generalization bounds, see e.g. [3]. As a final note, we highlight that Theorem 3.2 makes no specific assumption about the data (beyond requiring that $\lambda_F > 0$ so that the statement is meaningful, which holds for almost every dataset). In other settings, making additional assumptions on the data is crucial for obtaining further improvements [6, 12, 23, 30].

## 3.2 LeCun's initialization: Width $\Omega(N^2)$ suffices

The widely used LeCun's initialization, i.e., $[W_l^0]_{ij} \sim \mathcal{N}(0, 1/n_{l-1})$ for all $l \in [L], i \in [n_{l-1}], j \in [n_l]$, satisfies our Assumption 3.1 under a stronger requirement on the width of the first layer, and thus the results of Theorem 3.2 hold. For space reason, the formal statement and proof are postponed to Appendix C.3. There, our main Theorem C.4 applies to any training data. Below, we discuss how this result looks like when considering the following setting (standard in the literature): *(i)* $N \geq d$, *(ii)* the training samples lie on the sphere of radius $\sqrt{d}$, *(iii)* $n_L$ is a constant, and *(iv)* the target labels satisfy $\|y_i\| = \mathcal{O}(1)$ for $i \in [N]$. Then, Assumption 3.1 is satisfied w.h.p. if the first layer scales as:

$$n_1 = \Omega\left(\max\left(\frac{\|X\|_2^2}{\lambda_*}\left(\log\frac{N}{\lambda_*}\right)^2, \frac{N^2 2^{\mathcal{O}(L)}}{\lambda_*^2}\right)\right), \tag{12}$$

where $\lambda_*$ is the smallest eigenvalue of the expected Gram matrix w.r.t. the output of the first layer:

$$\lambda_* = \lambda_{\min}(G_*), \qquad G_* = \mathbb{E}_{w \sim \mathcal{N}(0, \frac{1}{d}\mathbb{I}_d)}\left[\sigma(Xw)\sigma(Xw)^T\right]. \tag{13}$$

Compared to Assumption 2.1, our result in this section also requires a slightly stronger requirement on the pyramidal topology, namely $\sqrt{n_{l-1}} \geq 1.01(\sqrt{n_l} + t)$, for some constant $t > 0$.

We note that the bound (12) holds for any training data that lie on the sphere. Now, let us discuss how this bound scales for *random data* (still on the sphere). First, we have that

$$\lambda_* \leq \frac{\text{tr}(G_*)}{N} = \frac{\mathbb{E}\|\sigma(Xw)\|_2^2}{N} \leq \frac{\mathbb{E}\|Xw\|_2^2}{N} = \frac{\|X\|_F^2}{Nd} = 1. \tag{14}$$

Then, if we additionally assume that the rows of $X \in \mathbb{R}^{N \times d}$ are sub-Gaussian and isotropic, $\|X\|_2^2$ is of order $N$, see e.g. Theorem 5.39 in [41]. As a result, we have that $\boxed{n_1 = \Omega(N^2/\lambda_*^2)}$.

To conclude, we briefly outline the steps leading to (12). First, by Gaussian concentration, we bound the output of the network at initialization (see Lemma C.1 in Appendix C.1). Then, we show that, with high probability, $\lambda_F \geq \sqrt{n_1 \lambda_*}/4$ (see Lemma C.2 in Appendix C.2). Finally, we bound the quantities $\bar{\lambda}_l$ and $\lambda_l$ using results on the singular values of random Gaussian matrices. By computing the terms $\alpha_0, Q_0$ and $Q_1$ defined in (6) and (8), we can also show that the number of iterations needed to achieve $\epsilon$ training loss scales as $\boxed{\dfrac{N^{3/2} 2^{\mathcal{O}(L)}}{\lambda_*} \log(1/\epsilon)}$. The next section shows that $\lambda_* = \Theta(1)$.

## 3.3 Lower bound on $\lambda_*$

By definition (13), $\lambda_*$ depends only on the activation $\sigma$ and on the training data $X$. Under some mild conditions on $(X, \sigma)$, one can show that $\lambda_* > 0$, see also the discussion at the end of Section 3.1. Nevertheless, this fact does not reveal how $\lambda_*$ scales with $N$ and $d$. Our next theorem shows that, for sub-Gaussian data, $\lambda_*$ is lower bounded by a constant independent of $N, d$. The detailed proof is given in Appendix D.4.

**Theorem 3.3** *Let $X = [x_1, \ldots, x_N]^T \in \mathbb{R}^{N \times d}$ be a matrix whose rows are i.i.d. random sub-Gaussian vectors with $\|x_i\|_2 = \sqrt{d}$ and $\|x_i\|_{\psi_2} \leq c_1$ for all $i \in [N]$, where $\|x_i\|_{\psi_2}$ denotes the sub-Gaussian norm of $x_i$ and $c_1$ is a numerical constant (independent of $d$). Assume that (i) $\sigma \in L^2(\mathbb{R}, e^{-x^2/2}/\sqrt{2\pi})$ [3], (ii) $\sigma$ is not a linear function, and (iii) $|\sigma(x)| \leq |x|$ for every $x \in \mathbb{R}$. Fix any integer $k \geq 2$. Then, for $N \leq d^k$, we have*

$$\mathbb{P}\left(\lambda_* \geq b_1\right) \geq 1 - 2N^2 e^{-c_2 N^{4/(5k)}}. \tag{15}$$

*Here, $c_2 > 0$ is independent of $(N, d, k)$, and $b_1 > 0$ is independent of $(N, d)$.*

We remark that the same result of Theorem 3.3 holds if $\|x_i\|_2 = r$ and $[W_1^0]_{ij} \sim \mathcal{N}(0, 1/r^2)$, for any $r > 0$. In order to handle different scalings of the data and the weights of the first layer, one would need to extend the Hermite analysis of Lemma D.3 in Appendix D.2.

By using (14), one immediately obtains that the lower bound (15) is tight (up to a constant). It is also necessary for $\sigma$ to be non-linear, otherwise $G_* = \frac{1}{d}XX^T$ and $\lambda_* = 0$ when $d < N$. Below we provide a proof sketch for Theorem 3.3. By using the Hermite expansion, one can show that

$$G_* = \sum_{r=0}^{\infty} \frac{\mu_r^2(\sigma)}{d^r} (X^{*r})(X^{*r})^T, \tag{16}$$

where $\mu_r(\sigma)$ denotes the $r$-th Hermite coefficient of $\sigma$, and each row of $X^{*r}$ is obtained by taking the Kronecker product of the corresponding row of $X$ with itself for $r$ times. The proof of (16) is given in Appendix D.2. As $\sigma$ is not a linear function and $|\sigma(x)| \leq |x|$ for $x \in \mathbb{R}$, we can show that $\mu_r(\sigma) > 0$ for arbitrarily large $r$. Thus, it remains to lower bound the smallest singular value of $X^{*r}$. This is done in the following lemma, whose proof appears in Appendix D.3.

**Lemma 3.4** *Let $X$ satisfy the assumptions of Theorem 3.3. Fix any integer $r \geq 2$. Then, for $N \leq d^r$, we have $\mathbb{P}\left(\sigma_{\min}(X^{*r}) \geq d^{r/2}/2\right) \geq 1 - 2N^2 e^{-c_3 dN^{-2/r}}$, where $c_3 > 0$ is independent of $(N, d, r)$.*

The probability in Lemma 3.4 tends to 1 as long as $N$ is $\mathcal{O}(d^{r/2-\epsilon})$ for any $\epsilon > 0$. We remark that it is possible to tighten this result for $r \in \{2, 3, 4\}$ (see Appendix D.5).

Lemma 3.4 also allows one to study the scaling of other quantities appearing in prior works. For instance, the $\lambda_0$ of Assumption 3.1 in [17] (see also Table 1) is $\Omega(1)$ when *(i)* the data points are sub-Gaussian and *(ii)* $N$ grows at most polynomially in $d$. If $N$ grows exponentially in $d$, then $\lambda_0$ is $\Omega((\log N)^{-3/2})$. This last estimate uses that the $r$-th Hermite coefficient of the step function scales as $1/r^{3/4}$. Similar considerations can be done for the bounds in [31].

Several prior works (see e.g. [2, 31, 48, 49]) derive bounds on the layer widths depending on the minimum distance between any pair of training data points. It would be interesting to see if such bounds can be improved by exploiting the techniques of this paper.

## 4 Proof of Theorem 3.2

Let $\otimes$ denote the Kronecker product, and let $\Sigma_l = \text{diag}[\text{vec}(\sigma'(G_l))] \in \mathbb{R}^{Nn_l \times Nn_l}$. Below, we frequently encounter situations where we need to evaluate the matrices $\Sigma_l, F_l$ at specific iterations of gradient descent. To this end, we define the shorthands $F_l^k = F_l(\theta_k), f_l^k = \text{vec}(F_l^k)$, and $\Sigma_l^k = \Sigma_l(\theta_k)$. We omit the parameter $\theta_k$ and write just $F_l, \Sigma_l$ when it is clear from the context. Now let us recall the following results from [28, 29], which provide a closed-form expression for the gradients, and a PL-type inequality for the training objective.

**Lemma 4.1** *Let Assumption 2.1 hold. Then, the following results hold:*

*1.* $\operatorname{vec}(\nabla_{W_l}\Phi) = (\mathbb{I}_{n_l} \otimes F_{l-1}^T) \prod_{p=l+1}^{L} \Sigma_{p-1}(W_p \otimes \mathbb{I}_N)(f_L - y).$

*2.* $\frac{\partial f_L}{\partial \operatorname{vec}(W_l)} = \prod_{p=0}^{L-l-1} (W_{L-p}^T \otimes \mathbb{I}_N)\Sigma_{L-p-1}(\mathbb{I}_{n_l} \otimes F_{l-1}).$

*3.* $\|\operatorname{vec}(\nabla_{W_2}\Phi)\|_2 \geq \sigma_{min}(F_1) \prod_{p=3}^{L} \sigma_{min}(\Sigma_{p-1})\,\sigma_{min}(W_p)\,\|f_L - y\|_2.$

The last statement of Lemma 4.1 requires the pyramidal network topology (see Assumption 2.1). In fact, the key idea of the proof (see Lemma 4.3 in [29]) is that the norm of the gradient can be lower bounded by the smallest singular value of $\prod_{p=3}^{L} A_p$ with $A_p = \Sigma_{p-1}(W_p \otimes \mathbb{I}_N) \in \mathbb{R}^{N\,n_{p-1} \times N\,n_p}$. Assuming that $n_2 \geq n_3 \geq \ldots \geq n_L$, one can further lower bound this quantity by the product of the smallest singular values of the $A_p$'s. This is where our assumption on the pyramidal topology comes from. Lemma 4.1 should be seen as providing a sufficient condition for a PL-inequality, rather than suggesting that such an inequality only holds for pyramidal networks.

The last statement of Lemma 4.1 implies the following fact: if $\sigma'(x) \neq 0$ for every $x \in \mathbb{R}$, then every stationary point $\theta = (W_l)_{l=1}^{L}$ for which $F_1$ has full rank and all the weight matrices $(W_l)_{l=3}^{L}$ have full rank is a global minimizer of $\Phi$. Consequently, in order to show convergence of GD to a global minimum, it suffices to *(i)* initialize all the matrices $\{F_1, W_3, \ldots, W_L\}$ to be full rank and *(ii)* make sure that the dynamics of GD stays inside the manifold of full-rank matrices. To do that, we show that the smallest singular value of those matrices stays bounded away from zero during training.

The following results are also required to show the main theorem (for their proofs, see Appendices B.2, B.3, B.4, B.5, B.6).

**Lemma 4.2** *Let Assumption 2.2 hold. Then, for every $\theta = (W_p)_{p=1}^{L}$ and $l \in [L]$,*

$$\|F_l\|_F \leq \|X\|_F \prod_{p=1}^{l} \|W_p\|_2, \tag{17}$$

$$\|\nabla_{W_l}\Phi\|_F \leq \|X\|_F \prod_{\substack{p=1 \\ p \neq l}}^{L} \|W_p\|_2 \, \|f_L - y\|_2. \tag{18}$$

*Furthermore, let $\theta_a = (W_l^a)_{l=1}^{L}, \theta_b = (W_l^b)_{l=1}^{L}$, and $\bar{\lambda}_l \geq \max(\|W_l^a\|_2, \|W_l^b\|_2)$ for some scalars $\bar{\lambda}_l$. Let $R = \prod_{p=1}^{L} \max(1, \bar{\lambda}_p)$. Then, for $l \in [L]$,*

$$\left\|F_L^a - F_L^b\right\|_F \leq \sqrt{L}\,\|X\|_F\, \frac{\prod_{l=1}^{L}\bar{\lambda}_l}{\min_{l\in[L]}\bar{\lambda}_l}\,\|\theta_a - \theta_b\|_2, \tag{19}$$

$$\left\|\frac{\partial f_L(\theta_a)}{\partial \operatorname{vec}(W_l^a)} - \frac{\partial f_L(\theta_b)}{\partial \operatorname{vec}(W_l^b)}\right\|_2 \leq \sqrt{L}\,\|X\|_F\, R\left(1 + L\beta\,\|X\|_F\, R\right)\|\theta_a - \theta_b\|_2. \tag{20}$$

**Lemma 4.3** *Let $f : \mathbb{R}^n \to \mathbb{R}$ be a $C^2$ function. Let $x, y \in \mathbb{R}^n$ be given, and assume that $\|\nabla f(z) - \nabla f(x)\|_2 \leq C\,\|z - x\|_2$ for every $z = x + t(y-x)$ with $t \in [0,1]$. Then,*

$$f(y) \leq f(x) + \langle \nabla f(x), y - x \rangle + \frac{C}{2}\,\|x - y\|^2.$$

At this point, we are ready to present the proof of our main result.

**Proof of Theorem 3.2.** We show by induction that, for every $k \geq 0$, the following holds:

$$\begin{cases}
\sigma_{\min}(W_l^r) \geq \frac{1}{2}\lambda_l, & l \in \{3, \ldots, L\},\, r \in \{0, \ldots, k\}, \\
\|W_l^r\|_2 \leq \frac{3}{2}\bar{\lambda}_l, & l \in \{1, \ldots, L\},\, r \in \{0, \ldots, k\}, \\
\sigma_{\min}(F_1^r) \geq \frac{1}{2}\lambda_F, & r \in \{0, \ldots, k\}, \\
\Phi(\theta_r) \leq (1 - \eta\alpha_0)^r \Phi(\theta_0), & r \in \{0, \ldots, k\},
\end{cases} \tag{21}$$

where $\lambda_l, \bar{\lambda}_l$ are defined in (3). Clearly, (21) holds for $k = 0$. Now, suppose that (21) holds for all iterations from 0 to $k$, and let us show the claim at iteration $k+1$. For every $r \in \{0, \ldots, k\}$, we have

$$\left\|W_l^{r+1} - W_l^0\right\|_F \leq \sum_{s=0}^{r} \left\|W_l^{s+1} - W_l^s\right\|_F = \eta \sum_{s=0}^{r} \left\|\nabla_{W_l} \Phi(\theta_s)\right\|_F$$

$$\leq \eta \sum_{s=0}^{r} \|X\|_F \prod_{\substack{p=1 \\ p \neq l}}^{L} \|W_p^s\|_2 \|f_L^s - y\|_2 \leq \eta \|X\|_F \left(\frac{3}{2}\right)^{L-1} \frac{\bar{\lambda}_{1 \to L}}{\bar{\lambda}_l} \sum_{s=0}^{r} (1 - \eta\alpha_0)^{s/2} \|f_L^0 - y\|_2,$$

where the 2nd inequality follows by (18), and the last one by induction hypothesis. Let $u := \sqrt{1 - \eta\alpha_0}$. Then, we can upper bound the RHS of the previous expression as

$$\frac{1}{\alpha_0} \|X\|_F \left(\frac{3}{2}\right)^{L-1} \frac{\bar{\lambda}_{1 \to L}}{\bar{\lambda}_l} (1 - u^2) \frac{1 - u^{r+1}}{1 - u} \|f_L^0 - y\|_2 \leq \begin{cases} \frac{1}{2}\lambda_l, & l \in \{3, \ldots, L\}, \\ 1, & l \in \{1, 2\}, \end{cases}$$

where the inequality follows from (4), definition of $\alpha_0$ and $u \in (0, 1)$. Thus by Weyl's inequality,

$$\begin{cases} \sigma_{\min}\left(W_l^{r+1}\right) \geq \sigma_{\min}\left(W_l^0\right) - \frac{1}{2}\lambda_l = \frac{1}{2}\lambda_l, & l \in \{3, \ldots, L\}, \\ \left\|W_l^{r+1}\right\|_2 \leq \left\|W_l^0\right\|_2 + \frac{1}{2}\bar{\lambda}_l = \frac{3}{2}\bar{\lambda}_l, & l \in \{3, \ldots, L\}, \\ \left\|W_1^{r+1}\right\|_2 \leq 1 + \left\|W_1^0\right\|_2 = \frac{3}{2}\bar{\lambda}_1, \\ \left\|W_2^{r+1}\right\|_2 \leq 1 + \left\|W_2^0\right\|_2 = \frac{3}{2}\bar{\lambda}_2. \end{cases}$$

Similarly, we have that, for every $r \in \{0, \ldots, k\}$

$$\left\|F_1^{r+1} - F_1^0\right\|_F = \left\|\sigma(XW_1^{r+1}) - \sigma(XW_1^0)\right\|_F \leq \|X\|_2 \left\|W_1^{r+1} - W_1^0\right\|_F$$

$$\leq \frac{2}{\alpha_0} \|X\|_2 \|X\|_F \left(\frac{3}{2}\right)^{L-1} \bar{\lambda}_{2 \to L} \|f_L^0 - y\|_2 \leq \frac{1}{2}\lambda_F,$$

where the first inequality uses Assumption 2.2, the second one follows from the above upper bound on $\left\|W_1^{r+1} - W_1^0\right\|_F$, and the last one uses (5). It follows that $\sigma_{\min}\left(F_1^{k+1}\right) \geq \sigma_{\min}\left(F_1^0\right) - \frac{1}{2}\lambda_F = \frac{1}{2}\lambda_F$. So far, we have shown that the first three statements in (21) hold for $k + 1$. It remains to show that $\Phi(\theta_{k+1}) \leq (1 - \eta\alpha_0)^{k+1}\Phi(\theta_0)$. To do so, define the shorthand $Jf_L$ for the Jacobian of the network: $Jf_L = \left[\frac{\partial f_L}{\partial \mathrm{vec}(W_1)}, \ldots, \frac{\partial f_L}{\partial \mathrm{vec}(W_L)}\right]$, where $\frac{\partial f_L}{\partial \mathrm{vec}(W_l)} \in \mathbb{R}^{(Nn_L) \times (n_{l-1}n_l)}$ for $l \in [L]$.

We first derive a Lipschitz constant for the gradient restricted to the line segment $[\theta_k, \theta_{k+1}]$. Let $\theta_k^t = \theta_k + t(\theta_{k+1} - \theta_k)$ for $t \in [0, 1]$. Then, by triangle inequality,

$$\left\|\nabla\Phi(\theta_k^t) - \nabla\Phi(\theta_k)\right\|_2 = \left\|Jf_L(\theta_k^t)^T [f_L(\theta_k^t) - y] - Jf_L(\theta_k)^T [f_L(\theta_k) - y]\right\|_2$$

$$\leq \left\|f_L(\theta_k^t) - f_L(\theta_k)\right\|_2 \left\|Jf_L(\theta_k^t)\right\|_2 + \left\|Jf_L(\theta_k^t) - Jf_L(\theta_k)\right\|_2 \|f_L(\theta_k) - y\|_2. \quad (22)$$

In the following, we bound each term in (22). We first note that, for $l \in [L]$ and $t \in [0, 1]$,

$$\left\|W_l(\theta_k^t) - W_l^0\right\|_F \leq \left\|W_l(\theta_k^t) - W_l^k\right\|_F + \sum_{s=0}^{k-1} \left\|W_l^{s+1} - W_l^s\right\|_F$$

$$= \left\|t\eta\nabla_{W_l}\Phi(\theta_k)\right\|_F + \sum_{s=0}^{k-1} \left\|\eta\nabla_{W_l}\Phi(\theta_s)\right\|_F \leq \eta \sum_{s=0}^{k} \left\|\nabla_{W_l}\Phi(\theta_s)\right\|_F.$$

By following a similar chain of inequalities as done in the beginning, we obtain that, for $l \in [L]$,

$$\max(\left\|W_l(\theta_k^t)\right\|_2, \|W_l(\theta_k)\|_2) \leq \frac{3}{2}\bar{\lambda}_l. \quad (23)$$

By using (19) and (23), we get

$$\left\|f_L(\theta_k^t) - f_L(\theta_k)\right\|_2 \leq \sqrt{L} \|X\|_F \left(\frac{3}{2}\right)^{L-1} \frac{\bar{\lambda}_{1 \to L}}{\min_{l \in [L]} \bar{\lambda}_l} \left\|\theta_k^t - \theta_k\right\|_2.$$

Note that, for a partitioned matrix $A = [A_1, \ldots, A_n]$, we have that $\|A\|_2 \leq \sum_{i=1}^{n} \|A_i\|_2$. Thus,

$$
\|Jf_L(\theta_k^t)\|_2 \leq \sum_{l=1}^{L} \left\| \frac{\partial f_L(\theta_k^t)}{\partial \operatorname{vec}(W_l)} \right\|_2 \leq \sum_{l=1}^{L} \prod_{p=l+1}^{L} \|W_p(\theta_k^t)\|_2 \|F_{l-1}(\theta_k^t)\|_2
$$

$$
\leq \|X\|_F \sum_{l=1}^{L} \prod_{\substack{p=1 \\ p \neq l}}^{L} \|W_p(\theta_k^t)\|_2 \leq \|X\|_F \left(\frac{3}{2}\right)^{L-1} \bar{\lambda}_{1 \to L} \sum_{l=1}^{L} \bar{\lambda}_l^{-1} \leq L \|X\|_F \left(\frac{3}{2}\right)^{L-1} \frac{\bar{\lambda}_{1 \to L}}{\min_{l \in [L]} \bar{\lambda}_l},
$$

where the second inequality follows by Lemma 4.1, the third by (17), the fourth by (23). Now we bound the Lipschitz constant of the Jacobian restricted to the segment $[\theta_k, \theta_{k+1}]$. From (20) and (23),

$$
\|Jf_L(\theta_k^t) - Jf_L(\theta_k)\|_2 \leq \sum_{l=1}^{L} \left\| \frac{\partial f_L(\theta_k^t)}{\operatorname{vec}(W_l)} - \frac{\partial f_L(\theta_k)}{\operatorname{vec}(W_l)} \right\|_2 \leq L^{3/2} \|X\|_F R \left(1 + L\beta \|X\|_F R\right) \|\theta_k^t - \theta_k\|_2.
$$

Plugging all these bounds into (22) gives $\|\nabla\Phi(\theta_k^t) - \nabla\Phi(\theta_k)\|_2 \leq Q_0 \|\theta_k^t - \theta_k\|_2$. By Lemma 4.3,

$$
\Phi(\theta_{k+1}) \leq \Phi(\theta_k) + \langle \nabla\Phi(\theta_k), \theta_{k+1} - \theta_k \rangle + \frac{Q_0}{2} \|\theta_{k+1} - \theta_k\|_2^2
$$

$$
= \Phi(\theta_k) - \eta \|\nabla\Phi(\theta_k)\|_2^2 + \frac{Q_0}{2}\eta^2 \|\nabla\Phi(\theta_k)\|_2^2
$$

$$
\leq \Phi(\theta_k) - \frac{1}{2}\eta \|\nabla\Phi(\theta_k)\|_2^2 \qquad \text{as } \eta < 1/Q_0
$$

$$
\leq \Phi(\theta_k) - \frac{1}{2}\eta \|\operatorname{vec}(\nabla_{W_2}\Phi(\theta_k))\|_2^2
$$

$$
\leq \Phi(\theta_k) - \frac{1}{2}\eta\, \sigma_{\min}\left(F_1^k\right)^2 \left[\prod_{p=3}^{L} \sigma_{\min}\left(\Sigma_{p-1}^k\right)^2 \sigma_{\min}\left(W_p^k\right)^2\right] \|f_L^k - y\|_2^2 \qquad \text{by Lemma 4.1,}
$$

$$
\leq \Phi(\theta_k) - \eta\, \gamma^{2(L-2)} \left(\frac{1}{2}\right)^{2(L-1)} \lambda_{3 \to L}^2 \lambda_F^2 \frac{1}{2} \|f_L^k - y\|_2^2 \qquad \text{by (21) and Assumption 2.2,}
$$

$$
= \Phi(\theta_k)(1 - \eta\alpha_0), \quad \text{by def. of } \alpha_0 \text{ in (6).}
$$

So far, we have proven the hypothesis (21). Using arguments similar to those at the beginning of this proof, one can show that $\{\theta_k\}_{k=0}^{\infty}$ is a Cauchy sequence and that (9) holds (a detailed proof is given in Appendix B.7). Thus, $\{\theta_k\}_{k=0}^{\infty}$ is a convergent sequence and there exists some $\theta_*$ such that $\lim_{k \to \infty} \theta_k = \theta_*$. By continuity, $\Phi(\theta_*) = \lim_{k \to \infty} \Phi(\theta_k) = 0$, hence $\theta_*$ is a global minimizer. $\quad\square$

## 5  Concluding Remarks

This paper shows that, for deep neural networks, a single layer of width $N$, where $N$ is the number of training samples, suffices to guarantee linear convergence of gradient descent to a global optimum. All the remaining layers are allowed to have constant widths and form a pyramidal topology. This result complements the previous loss surface analysis [26, 28, 29] by providing the missing algorithmic guarantee. We regard as an open question to understand the generalization properties of deep pyramidal networks. Other two interesting directions arising from our work are as follows:

- Can we trade off larger depth for smaller width at the first layer, while maintaining the pyramidal topology for the top layers?

- Can we extend our analysis of networks with "one wide layer" to ReLU activations? Our approach is currently not suitable since *(i)* the derivative of ReLU is not Lipschitz, which is needed to prove (20), and *(ii)* ReLU can have zero derivative, while we need $\gamma > 0$ for the PL-inequality to hold. The second problem seems to be more fundamental, i.e. how to show a PL-inequality for ReLU and ensure that it holds throughout the trajectory of GD.

## Broader Impact

This work does not present any foreseeable societal consequence.

## Acknowledgements

The authors would like to thank Jan Maas, Mahdi Soltanolkotabi, and Daniel Soudry for the helpful discussions, Marius Kloft, Matthias Hein and Quoc Dinh Tran for proofreading portions of a prior version of this paper, and James Martens for a clarification concerning LeCun's initialization. M. Mondelli was partially supported by the 2019 Lopez-Loreta Prize. Q. Nguyen was partially supported by the German Research Foundation (DFG) award KL 2698/2-1.

## Footnotes

[2]For analytic activation functions such as (2), and almost all training data, the set of $W_1^0$ for which $\sigma(XW_1^0)$ does not have full rank has measure zero.

[3] The condition $\sigma \in L^2(\mathbb{R}, e^{-x^2/2}/\sqrt{2\pi})$ means that $\int_{\mathbb{R}} \frac{1}{\sqrt{2\pi}} |\sigma(x)|^2 e^{-x^2/2} dx < \infty$. One can easily check that most of the popular activation functions in deep learning is contained in this $L^2$ space, including (2).

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
