[Supplementary Material]

# Supplementary Material (Appendix)

# Global Convergence of Deep Networks with One Wide Layer Followed by Pyramidal Topology

## A  Mathematical Tools

**Proposition A.1 (Weyl's inequality, see e.g. [40])** *Let $A, B \in \mathbb{R}^{m \times n}$ with $\sigma_1(A) \geq \ldots \geq \sigma_r(A)$ and $\sigma_1(B) \geq \ldots \geq \sigma_r(B)$, where $r = \min(m, n)$. Then,*

$$\max_{i \in [r]} |\sigma_i(A) - \sigma_i(B)| \leq \|A - B\|_2.$$

**Lemma A.2 (Singular values of random gaussian matrices, see e.g. [41])** *Let $A \in \mathbb{R}^{m \times n}$ be a random matrix with $m \geq n$ and $A_{ij} \overset{iid}{\sim} \mathcal{N}(0, 1)$. For every $t \geq 0$, it holds w.p. $\geq 1 - 2e^{-t^2/2}$*

$$\sqrt{m} - \sqrt{n} - t \leq \sigma_{min}(A) \leq \|A\|_2 \leq \sqrt{m} + \sqrt{n} + t.$$

**Theorem A.3 (Matrix Chernoff)** *Let $\{X_i\}_{i=1}^n \in \mathbb{R}^{d \times d}$ be a sequence of independent, random, symmetric matrices. Assume that $0 \leq \lambda_{min}(X_i) \leq \lambda_{max}(X_i) \leq R$. Let $S = \sum_{i=1}^n X_i$. Then,*

$$\mathbb{P}\left(\lambda_{min}(S) \leq (1 - \epsilon)\lambda_{min}(\mathbb{E}S)\right) \leq d \left[\frac{e^{-\epsilon}}{(1 - \epsilon)^{1-\epsilon}}\right]^{\lambda_{min}(\mathbb{E}S)/R} \quad \forall \epsilon \in [0, 1),$$

$$\mathbb{P}\left(\lambda_{max}(S) \geq (1 + \epsilon)\lambda_{max}(\mathbb{E}S)\right) \leq d \left[\frac{e^{\epsilon}}{(1 + \epsilon)^{1+\epsilon}}\right]^{\lambda_{max}(\mathbb{E}S)/R} \quad \forall \epsilon \geq 0.$$

## B  Proofs for General Framework (Theorem 3.2)

In the following, we frequently use a basic inequality, namely, for every $A, B \in \mathbb{R}^{m \times n}$, $\|AB\|_F \leq \|A\|_2 \|B\|_F$ and $\|AB\|_F \leq \|A\|_F \|B\|_2$.

### B.1  Properties of Activation Function (2)

**Lemma B.1** *Let $\sigma : \mathbb{R} \to \mathbb{R}$ be given as in (2). Then,*

1. *$\sigma$ is real analytic.*

2. *$\sigma'(x) \in [\gamma, 1]$ for every $x \in \mathbb{R}$.*

3. *$|\sigma(x)| \leq |x|$ for every $x \in \mathbb{R}$.*

4. *$\sigma'$ is $\beta$-Lipschitz.*

5. *$\lim_{\beta \to \infty} \sup_{x \in \mathbb{R}} |\sigma(x) - \max(\gamma x, x)| = 0.$*

**Proof:** Let $\Psi$ be the CDF of the standard normal distribution. Then, after some manipulations, we have that

$$\sigma(x) = -\frac{(1-\gamma)^2}{2\pi\beta} + \frac{(1-\gamma)^2}{2\pi\beta} \exp\left(-\frac{\pi\beta^2 x^2}{(1-\gamma)^2}\right) + x\Psi\left(\frac{\beta\sqrt{2\pi}x}{1-\gamma}\right) + \gamma x\Psi\left(-\frac{\beta\sqrt{2\pi}x}{1-\gamma}\right). \tag{24}$$

1. Since $\Psi$ is known as an entire function (i.e. analytic everywhere), it follows from (24) that $\sigma$ is analytic on $\mathbb{R}$.

2. Note that $\Psi'(z) = \frac{1}{\sqrt{2\pi}} e^{-z^2/2}$ and $\Psi(-z) = 1 - \Psi(z)$. Thus, after some simplifications, we have that

$$\sigma'(x) = \gamma + (1 - \gamma)\Psi\left(\frac{\beta\sqrt{2\pi}x}{1 - \gamma}\right). \tag{25}$$

The result follows by noting that $\Psi(\cdot) \in [0, 1]$.

3. It is easy to check that $\sigma(0) = 0$. Moreover, $\sigma$ is 1-Lipschitz and thus $|\sigma(x)| = |\sigma(x) - \sigma(0)| \leq |x|$.

4. We have that

$$\sigma''(x) = \beta\sqrt{2\pi}\Psi'\left(\frac{\beta\sqrt{2\pi}x}{1 - \gamma}\right) \leq \beta.$$

Thus $\sigma'$ is $\beta$-Lipschitz.

5. Note that

$$\frac{\beta}{1 - \gamma} \int_{-\infty}^{\infty} \exp\left(-\frac{\pi\beta^2(x - u)^2}{(1 - \gamma)^2}\right) du = 1,$$

which implies that

$$\max(\gamma x, x) = \frac{\beta}{1 - \gamma} \int_{-\infty}^{\infty} \max(\gamma x, x) \exp\left(-\frac{\pi\beta^2(x - u)^2}{(1 - \gamma)^2}\right) du.$$

Thus, the following chain of inequalities holds:

$$|\sigma(x) - \max(\gamma x, x)|$$

$$= \left| -\frac{(1 - \gamma)^2}{2\pi\beta} + \frac{\beta}{1 - \gamma} \int_{-\infty}^{\infty} \max(\gamma u, u) \exp\left(-\frac{\pi\beta^2(x - u)^2}{(1 - \gamma)^2)}\right) du \right.$$

$$\left. -\frac{\beta}{1 - \gamma} \int \max(\gamma x, x) \exp\left(-\frac{\pi\beta^2(x - u)^2}{(1 - \gamma)^2}\right) du \right|$$

$$\leq \frac{(1 - \gamma)^2}{2\pi\beta} + \frac{\beta}{1 - \gamma} \int_{-\infty}^{\infty} |\max(\gamma u, u) - \max(\gamma x, x)| \exp\left(-\frac{\pi\beta^2(x - u)^2}{(1 - \gamma)^2}\right) du$$

$$\leq \frac{(1 - \gamma)^2}{2\pi\beta} + \frac{\beta}{1 - \gamma} \int_{-\infty}^{\infty} |x - u| \exp\left(-\frac{\pi\beta^2(x - u)^2}{(1 - \gamma)^2}\right) du$$

$$= \frac{(1 - \gamma)^2}{2\pi\beta} + \frac{\beta}{1 - \gamma} \int_{-\infty}^{\infty} |v| \exp\left(-\frac{\pi\beta^2 v^2}{(1 - \gamma)^2}\right) dv$$

$$= \frac{(1 - \gamma)^2}{2\pi\beta} + 2\frac{\beta}{1 - \gamma} \int_{0}^{\infty} v \exp\left(-\frac{\pi\beta^2 v^2}{(1 - \gamma)^2}\right) dv$$

$$= \frac{(1 - \gamma)^2}{2\pi\beta} + \frac{1 - \gamma}{\pi\beta}.$$

Taking the supremum and the limit on both sides yields the result.

$\square$

## B.2  Proof of (17) in Lemma 4.2

We prove by induction on $l$. Note that the lemma holds for $l = 1$ since

$$\|F_1\|_F = \|\sigma(XW_1)\|_F \leq \|XW_1\|_F \leq \|X\|_F \|W_1\|_2,$$

where in the 2nd step we use our assumption on $\sigma$. Assume the lemma holds for $l - 1$, i.e.

$$\|F_{l-1}\|_F \leq \|X\|_F \prod_{p=1}^{l-1} \|W_p\|_2.$$

It is easy to verify that it also holds for $l$. Indeed,

$$
\begin{aligned}
\|F_l\|_F &= \|\sigma(F_{l-1}W_l)\|_F && \text{by definition} \\
&\leq \|F_{l-1}W_l\|_F && |\sigma(x)| \leq |x| \\
&\leq \|F_{l-1}\|_F \|W_l\|_2 \\
&\leq \|X\|_F \prod_{p=1}^{l} \|W_p\|_2 && \text{by induction assump.}
\end{aligned}
$$

For $l = L$ one can skip the first equality above, as there is no activation at the output layer. $\qquad\square$

### B.3   Proof of (19) in Lemma 4.2

We first prove the following intermediate result.

**Lemma B.2** *Let $\sigma$ be 1-Lipschitz and $|\sigma(x)| \leq |x|$ for every $x \in \mathbb{R}$. Let $\theta_a = (W_l^a)_{l=1}^L, \theta_b = (W_l^b)_{l=1}^L$. Let $\bar{\lambda}_l \geq \max(\|W_l^a\|_2, \|W_l^b\|_2)$. Then, for every $l \in [L]$,*

$$
\begin{aligned}
\left\|F_l^a - F_l^b\right\|_F &\leq \left\|G_l^a - G_l^b\right\|_F \\
&\leq \|X\|_F \, \bar{\lambda}_{1\to l} \sum_{p=1}^{l} \bar{\lambda}_p^{-1} \left\|W_p^a - W_p^b\right\|_2.
\end{aligned}
$$

*Here, we denote $\bar{\lambda}_{i\to j} = \prod_{l=i}^{j} \bar{\lambda}_l$.*

**Proof:** We prove by induction on $l$. First, it holds for $l = 1$ since

$$
\begin{aligned}
\left\|F_1^a - F_1^b\right\|_F &= \left\|\sigma(G_1^a) - \sigma(G_1^b)\right\|_F && \text{by definition} \\
&\leq \left\|G_1^a - G_1^b\right\|_F && \sigma \text{ is 1-Lipschitz} \\
&= \left\|XW_1^a - XW_1^b\right\|_F \\
&\leq \|X\|_F \left\|W_1^a - W_1^b\right\|_2.
\end{aligned}
$$

Suppose the lemma holds for $l - 1$ and we want to prove it for $l$. We have

$$
\begin{aligned}
\left\|F_l^a - F_l^b\right\|_F &= \left\|\sigma(G_l^a) - \sigma(G_l^b)\right\|_F && \text{definition} \\
&\leq \left\|G_l^a - G^b\right\|_F && \sigma \text{ is 1-Lipschitz} \\
&= \left\|F_{l-1}^a W_l^a - F_{l-1}^b W_l^b\right\|_F \\
&\leq \left\|F_{l-1}^a W_l^a - F_{l-1}^b W_l^a\right\|_F + \left\|F_{l-1}^b W_l^a - F_{l-1}^b W_l^b\right\|_F && \text{triangle inequality} \\
&\leq \left\|F_{l-1}^a - F_{l-1}^b\right\|_F \|W_l^a\|_2 + \left\|F_{l-1}^b\right\|_F \left\|W_l^a - W_l^b\right\|_2 \\
&\leq \left\|F_{l-1}^a - F_{l-1}^b\right\|_F \|W_l^a\|_2 + \|X\|_F \left[\prod_{p=1}^{l-1} \|W_p^b\|_2\right] \left\|W_l^a - W_l^b\right\|_2 && \text{by (17)} \\
&\leq \left\|F_{l-1}^a - F_{l-1}^b\right\|_F \bar{\lambda}_l + \|X\|_F \, \bar{\lambda}_{1\to l-1} \left\|W_l^a - W_l^b\right\|_2 \\
&\leq \|X\|_F \, \bar{\lambda}_{1\to l} \sum_{p=1}^{l} \bar{\lambda}_p^{-1} \left\|W_p^a - W_p^b\right\|_2 && \text{induction assumption}
\end{aligned}
$$

$\qquad\square$

Applying Lemma B.2 to the output layer yields:

$$
\begin{aligned}
\left\|F_L^a - F_L^b\right\|_F &= \left\|G_L^a - G_L^b\right\|_F \\
&\leq \|X\|_F \, \bar{\lambda}_{1\to L} \sum_{p=1}^{L} \bar{\lambda}_p^{-1} \left\|W_p^a - W_p^b\right\|_2 \\
&\leq \sqrt{L}\, \|X\|_F \frac{\bar{\lambda}_{1\to L}}{\min_{l\in[L]} \bar{\lambda}_l} \|\theta_a - \theta_b\|_2 && \text{Cauchy-Schwarz}
\end{aligned}
$$

$\qquad\square$

## B.4 Proof of (18) in Lemma 4.2

$$\|\nabla_{W_l}\Phi\|_F = \|\text{vec}(\nabla_{W_l}\Phi)\|_2$$

$$= \left\| (\mathbb{I}_{n_l} \otimes F_{l-1}^T) \left[ \prod_{p=l+1}^{L} \Sigma_{p-1}(W_p \otimes \mathbb{I}_N) \right] (f_L - y) \right\|_2 \qquad \text{Lemma 4.1}$$

$$\leq \|F_{l-1}\|_2 \left[ \prod_{p=l+1}^{L} \|W_p\|_2 \right] \|f_L - y\|_2 \qquad |\sigma'| \leq 1$$

$$\leq \|X\|_F \left[ \prod_{\substack{p=1 \\ p \neq l}}^{L} \|W_p^k\|_2 \right] \|f_L - y\|_2 \qquad \text{by (17).}$$

$\square$

## B.5 Proof of (20) in Lemma 4.2

We start by showing the following intermediate result.

**Lemma B.3** *Let $\sigma$ be 1-Lipschitz, and let $|\sigma(x)| \leq |x|$ and $|\sigma'(x)| \leq 1$ hold for every $x \in \mathbb{R}$. Let $\theta_a = (W_l^a)_{l=1}^L, \theta_b = (W_l^a)_{l=1}^L$. Let $\bar{\lambda}_l \geq \max(\|W_l^a\|_2, \|W_l^b\|_2)$. Then, for every $l \in [L]$,*

$$\left\| \frac{\partial f_L(\theta_a)}{\partial \text{vec}(W_l^a)} - \frac{\partial f_L(\theta_b)}{\partial \text{vec}(W_l^b)} \right\|_2 \leq \|X\|_F \, \bar{\lambda}_{1 \to L} \bar{\lambda}_l^{-1} \sum_{p=l+1}^{L} \bar{\lambda}_p^{-1} \|W_p^a - W_p^b\|_2$$

$$+ \|X\|_F \, \bar{\lambda}_{1 \to L} \bar{\lambda}_l^{-1} \sum_{p=l}^{L-1} \|\Sigma_p^a - \Sigma_p^b\|_2 + \bar{\lambda}_{l+1 \to L} \|F_{l-1}^a - F_{l-1}^b\|_2.$$

*Here, we denote $\bar{\lambda}_{i \to j} = \prod_{l=i}^{j} \bar{\lambda}_l$.*

**Proof:** For every $t \in \{l, \ldots, L\}$, let

$$M_t^a = \left[ \prod_{p=t \to l+1} ((W_p^a)^T \otimes \mathbb{I}_N) \Sigma_{p-1}^a \right] (\mathbb{I}_{n_l} \otimes F_{l-1}^a),$$

$$M_t^b = \left[ \prod_{p=t \to l+1} ((W_p^b)^T \otimes \mathbb{I}_N) \Sigma_{p-1}^b \right] (\mathbb{I}_{n_l} \otimes F_{l-1}^b).$$

In the above definition, we note that $p$ runs in the reverse order, that is, $p = t, t-1, \ldots, l+1$. For the case $t = l$ (the terms inside brackets are inactive), we assume by convention that $M_l^a = (\mathbb{I}_{n_l} \otimes F_{l-1}^a)$ and $M_l^b = (\mathbb{I}_{n_l} \otimes F_{l-1}^b)$. It follows from Lemma 4.1 that

$$\frac{\partial f_L(\theta_a)}{\partial \text{vec}(W_l)} = M_L^a, \qquad \frac{\partial f_L(\theta_b)}{\partial \text{vec}(W_l)} = M_L^b.$$

The following inequality holds

$$\|M_t^a\|_2 \leq \left[ \prod_{p=l+1}^{t} \|W_p^a\|_2 \|\Sigma_{p-1}^a\|_2 \right] \|F_{l-1}^a\|_2$$

$$\leq \left[ \prod_{p=l+1}^{t} \|W_p^a\|_2 \right] \|X\|_F \left[ \prod_{p=1}^{l-1} \|W_p^a\|_2 \right]$$

$$\leq \bar{\lambda}_{1 \to t} \bar{\lambda}_l^{-1} \|X\|_F, \qquad (26)$$

where the second inequality follows from (17) and $|\sigma'| \leq 1$. To prove the lemma, we will prove that, for every $t \in \{l, \ldots, L\}$,

$$
\left\| M_t^a - M_t^b \right\|_2 \leq \|X\|_F \sum_{p=l+1}^{t} \bar{\lambda}_{1 \to t} \bar{\lambda}_p^{-1} \bar{\lambda}_l^{-1} \left\| W_p^a - W_p^b \right\|_2
$$

$$
+ \|X\|_F \, \bar{\lambda}_{1 \to t} \bar{\lambda}_l^{-1} \sum_{p=l}^{t-1} \left\| \Sigma_p^a - \Sigma_p^b \right\|_2 + \bar{\lambda}_{l+1 \to t} \left\| F_{l-1}^a - F_{l-1}^b \right\|_2. \tag{27}
$$

Then setting $t = L$ in (27) leads to the desired result. First we note that (27) holds for $t = l$ since

$$
\left\| M_l^a - M_l^b \right\|_2 = \left\| (\mathbb{I}_{n_l} \otimes F_{l-1}^a) - (\mathbb{I}_{n_l} \otimes F_{l-1}^b) \right\|_2 = \left\| F_{l-1}^a - F_{l-1}^b \right\|_2.
$$

Suppose that it holds for $t - 1$ with $t \geq l + 1$, and we want to show it for $t$. Then,

$$
\left\| M_t^a - M_t^b \right\|_2 = \left\| ((W_t^a)^T \otimes \mathbb{I}_N) \Sigma_{t-1}^a M_{t-1}^a - ((W_t^b)^T \otimes \mathbb{I}_N) \Sigma_{t-1}^b M_{t-1}^b \right\|_2
$$

$$
\leq \left\| ((W_t^a)^T \otimes \mathbb{I}_N) \Sigma_{t-1}^a M_{t-1}^a - ((W_t^b)^T \otimes \mathbb{I}_N) \Sigma_{t-1}^a M_{t-1}^a \right\|_2
$$

$$
+ \left\| ((W_t^b)^T \otimes \mathbb{I}_N) \Sigma_{t-1}^a M_{t-1}^a - ((W_t^b)^T \otimes \mathbb{I}_N) \Sigma_{t-1}^b M_{t-1}^b \right\|_2
$$

$$
\leq \left\| W_t^a - W_t^b \right\|_2 \left\| \Sigma_{t-1}^a \right\|_2 \left\| M_{t-1}^a \right\|_2 + \left\| W_t^b \right\|_2 \left\| \Sigma_{t-1}^a M_{t-1}^a - \Sigma_{t-1}^b M_{t-1}^b \right\|_2
$$

$$
\leq \left\| W_t^a - W_t^b \right\|_2 \bar{\lambda}_{1 \to t-1} \bar{\lambda}_l^{-1} \|X\|_F + \bar{\lambda}_t \left\| \Sigma_{t-1}^a M_{t-1}^a - \Sigma_{t-1}^b M_{t-1}^b \right\|_2, \qquad \text{by (26) and } |\sigma'| \leq 1
$$

$$
\leq \left\| W_t^a - W_t^b \right\|_2 \bar{\lambda}_{1 \to t-1} \bar{\lambda}_l^{-1} \|X\|_F
$$

$$
+ \bar{\lambda}_t \left[ \left\| \Sigma_{t-1}^a M_{t-1}^a - \Sigma_{t-1}^b M_{t-1}^a \right\|_2 + \left\| \Sigma_{t-1}^b M_{t-1}^a - \Sigma_{t-1}^b M_{t-1}^b \right\|_2 \right]
$$

$$
\leq \left\| W_t^a - W_t^b \right\|_2 \bar{\lambda}_{1 \to t-1} \bar{\lambda}_l^{-1} \|X\|_F
$$

$$
+ \bar{\lambda}_t \left[ \left\| \Sigma_{t-1}^a - \Sigma_{t-1}^b \right\|_2 \bar{\lambda}_{1 \to t-1} \bar{\lambda}_l^{-1} \|X\|_F + \left\| M_{t-1}^a - M_{t-1}^b \right\|_2 \right]
$$

$$
= \|X\|_F \, \bar{\lambda}_{1 \to t-1} \bar{\lambda}_l^{-1} \left\| W_t^a - W_t^b \right\|_2 + \|X\|_F \, \bar{\lambda}_{1 \to t} \bar{\lambda}_l^{-1} \left\| \Sigma_{t-1}^a - \Sigma_{t-1}^b \right\|_2 + \bar{\lambda}_t \left\| M_{t-1}^a - M_{t-1}^b \right\|_2
$$

$$
\leq \|X\|_F \, \bar{\lambda}_{1 \to t} \bar{\lambda}_l^{-1} \sum_{p=l+1}^{t} \bar{\lambda}_p^{-1} \left\| W_p^a - W_p^b \right\|_2
$$

$$
+ \|X\|_F \, \bar{\lambda}_{1 \to t} \bar{\lambda}_l^{-1} \sum_{p=l}^{t-1} \left\| \Sigma_p^a - \Sigma_p^b \right\|_2 + \bar{\lambda}_{l+1 \to t} \left\| F_{l-1}^a - F_{l-1}^b \right\|_2,
$$

where the last line follows by plugging the bound of $\left\| M_{t-1}^a - M_{t-1}^b \right\|_2$ from the induction assumption. $\qquad \square$

**Proof of** (20) **in Lemma 4.2.** Let

$$
S = \|X\|_F \, \bar{\lambda}_{1 \to L} \bar{\lambda}_l^{-1} \sum_{p=l+1}^{L} \bar{\lambda}_p^{-1} \left\| W_p^a - W_p^b \right\|_2.
$$

Then, by Lemma B.3, we have that

$$
\left\| \frac{\partial f_L(\theta_a)}{\text{vec}(W_l^a)} - \frac{\partial f_L(\theta_b)}{\text{vec}(W_l^b)} \right\|_2 \leq S + \|X\|_F \, \bar{\lambda}_{1 \to L} \bar{\lambda}_l^{-1} \sum_{p=l}^{L-1} \left\| \Sigma_p^a - \Sigma_p^b \right\|_2 + \bar{\lambda}_{l+1 \to L} \left\| F_{l-1}^a - F_{l-1}^b \right\|_2
$$

$$
= S + \|X\|_F \, \bar{\lambda}_{1 \to L} \bar{\lambda}_l^{-1} \sum_{p=l}^{L-1} \left\| \sigma'(G_p^a) - \sigma'(G_p^b) \right\|_2 + \bar{\lambda}_{l+1 \to L} \left\| F_{l-1}^a - F_{l-1}^b \right\|_2.
$$

$$
\tag{28}
$$

Furthermore, by using that $\sigma'$ is $\beta$-Lipschitz, the RHS of (28) is upper bounded by

$$
S + \|X\|_F \, \bar{\lambda}_{1 \to L} \bar{\lambda}_l^{-1} \sum_{p=l}^{L-1} \beta \left\| G_p^a - G_p^b \right\|_2 + \bar{\lambda}_{l+1 \to L} \left\| F_{l-1}^a - F_{l-1}^b \right\|_2. \tag{29}
$$

By applying Lemma B.2, the following chain of upper bounds for (29) holds:

$$S + \|X\|_F \, \bar{\lambda}_{1 \to L} \bar{\lambda}_l^{-1} \sum_{p=l}^{L-1} \beta \, \|X\|_F \, \bar{\lambda}_{1 \to p} \sum_{q=1}^{p} \bar{\lambda}_q^{-1} \left\| W_q^a - W_q^b \right\|_2$$

$$+ \, \bar{\lambda}_{l+1 \to L} \, \|X\|_F \, \bar{\lambda}_{1 \to l-1} \sum_{p=1}^{l-1} \bar{\lambda}_p^{-1} \left\| W_p^a - W_p^b \right\|_2$$

$$= \|X\|_F^2 \, \beta \bar{\lambda}_{1 \to L} \bar{\lambda}_l^{-1} \sum_{p=l}^{L-1} \bar{\lambda}_{1 \to p} \sum_{q=1}^{p} \bar{\lambda}_q^{-1} \left\| W_q^a - W_q^b \right\|_2$$

$$+ \, \|X\|_F \, \bar{\lambda}_{1 \to L} \bar{\lambda}_l^{-1} \sum_{\substack{p=1 \\ p \neq l}}^{L} \bar{\lambda}_p^{-1} \left\| W_p^a - W_p^b \right\|_2$$

$$\leq \|X\|_F^2 \, \beta \bar{\lambda}_{1 \to L} \bar{\lambda}_l^{-1} \sum_{p=1}^{L} \left[ \prod_{q=1}^{L} \max(1, \bar{\lambda}_q) \right] \sum_{q=1}^{p} \left\| W_q^a - W_q^b \right\|_2 \qquad (30)$$

$$+ \, \|X\|_F \left[ \prod_{p=1}^{L} \max(1, \bar{\lambda}_p) \right] \sum_{p=1}^{L} \left\| W_p^a - W_p^b \right\|_2$$

$$\leq L\beta \, \|X\|_F^2 \left[ \prod_{q=1}^{L} \max(1, \bar{\lambda}_q) \right]^2 \sum_{q=1}^{L} \left\| W_q^a - W_q^b \right\|_2$$

$$+ \, \|X\|_F \left[ \prod_{p=1}^{L} \max(1, \bar{\lambda}_p) \right] \sum_{p=1}^{L} \left\| W_p^a - W_p^b \right\|_2$$

$$= \|X\|_F \, R(1 + L\beta \, \|X\|_F \, R) \sum_{q=1}^{L} \left\| W_q^a - W_q^b \right\|_2$$

$$\leq \sqrt{L} \, \|X\|_F \, R(1 + L\beta \, \|X\|_F \, R) \sum_{q=1}^{L} \left\| \theta_a - \theta_b \right\|_2 \, ,$$

where the last passage follows from Cauchy-Schwarz inequality. By combining (28), (29) and (30), the result immediately follows. $\qquad \square$

## B.6   Proof of Lemma 4.3

Let $g(t) = f(x + t(y - x))$. Then

$$f(y) - f(x) = g(1) - g(0) = \int_0^1 g'(t)dt$$

$$= \int_0^1 \langle \nabla f(x + t(y - x)), y - x \rangle \, dt$$

$$= \langle \nabla f(x), y - x \rangle + \int_0^1 \langle \nabla f(x + t(y - x)) - \nabla f(x), y - x \rangle \, dt$$

$$\leq \langle \nabla f(x), y - x \rangle + \int_0^1 Ct \, \|y - x\|_2^2 \, dt$$

$$= \langle \nabla f(x), y - x \rangle + \frac{C}{2} \, \|x - y\|^2 \, .$$

$\qquad \square$

## B.7 Proof of the fact that $\{\theta_k\}_{k=1}^{\infty}$ is a Cauchy Sequence

Let us fix any $\epsilon > 0$. We need to show that there exists $r > 0$ such that for every $i, j \geq r$, $\|\theta_j - \theta_i\| < \epsilon$. The case $i = j$ is trivial, so we assume w.l.o.g. that $i < j$. Then, the following chain of inequalities hold

$$\|\theta_j - \theta_i\| = \sqrt{\sum_{l=1}^{L} \left\| W_l^j - W_l^i \right\|_F^2}$$

$$\leq \sum_{l=1}^{L} \left\| W_l^j - W_l^i \right\|_F$$

$$\leq \sum_{l=1}^{L} \sum_{s=i}^{j-1} \left\| W_l^{s+1} - W_l^s \right\|_F \qquad \text{triangle inequality}$$

$$= \sum_{l=1}^{L} \sum_{s=i}^{j-1} \eta \left\| \nabla_{W_l} \Phi(\theta_s) \right\|_F$$

$$\leq \sum_{l=1}^{L} \sum_{s=i}^{j-1} \eta \left\| X \right\|_F \left\| f_L^s - y \right\|_2 \prod_{\substack{p=1 \\ p \neq l}}^{L} \left\| W_p^s \right\|_2 \qquad \text{by (18)}$$

$$\leq \sum_{l=1}^{L} \eta \left\| X \right\|_F 1.5^{L-1} \bar{\lambda}_l^{-1} \bar{\lambda}_{1 \to L} \sum_{s=i}^{j-1} (1 - \eta\alpha_0)^{s/2} \left\| f_L^0 - y \right\|_2 \qquad \text{by (21)}$$

$$= (1 - \eta\alpha_0)^{i/2} \left[ \sum_{l=1}^{L} \eta \left\| X \right\|_F 1.5^{L-1} \bar{\lambda}_l^{-1} \bar{\lambda}_{1 \to L} \sum_{s=0}^{j-i-1} (1 - \eta\alpha_0)^{s/2} \left\| f_L^0 - y \right\|_2 \right]$$

$$= (1 - \eta\alpha_0)^{i/2} \left[ \eta \left\| X \right\|_F 1.5^{L-1} \sum_{l=1}^{L} \bar{\lambda}_l^{-1} \bar{\lambda}_{1 \to L} \frac{1 - \sqrt{1 - \eta\alpha_0}^{j-i}}{1 - \sqrt{1 - \eta\alpha_0}} \left\| f_L^0 - y \right\|_2 \right]$$

$$= (1 - \eta\alpha_0)^{i/2} \left[ \frac{1}{\alpha_0} \left\| X \right\|_F 1.5^{L-1} \sum_{l=1}^{L} \bar{\lambda}_l^{-1} \bar{\lambda}_{1 \to L} (1 - u^2) \frac{1 - u^{j-i}}{1 - u} \left\| f_L^0 - y \right\|_2 \right],$$

where we have set $u := \sqrt{1 - \eta\alpha_0}$. As $u \in (0, 1)$, the last term is upper bounded by

$$(1 - \eta\alpha_0)^{i/2} \left[ \frac{2}{\alpha_0} \left\| X \right\|_F 1.5^{L-1} \sum_{l=1}^{L} \bar{\lambda}_l^{-1} \bar{\lambda}_{1 \to L} \left\| f_L^0 - y \right\|_2 \right].$$

Note that $(1 - \eta\alpha_0)^{i/2} \leq (1 - \eta\alpha_0)^{r/2}$ and thus there exists a sufficiently large $r$ such that $\|\theta_j - \theta_i\| < \epsilon$. This shows that $\{\theta_k\}_{k=0}^{\infty}$ is a Cauchy sequence, and hence convergent to some $\theta_*$. By continuity, $\Phi(\theta_*) = \Phi(\lim_{k \to \infty} \theta_k) = \lim_{k \to \infty} \Phi(\theta_k) = 0$, and thus $\theta_*$ is a global minimizer. The rate of convergence is

$$\|\theta_k - \theta_*\| = \lim_{j \to \infty} \|\theta_k - \theta_j\| \leq (1 - \eta\alpha_0)^{k/2} \left[ \frac{2}{\alpha_0} \left\| X \right\|_F 1.5^{L-1} \sum_{l=1}^{L} \bar{\lambda}_l^{-1} \bar{\lambda}_{1 \to L} \left\| f_L^0 - y \right\|_2 \right].$$

$\square$

## C Proofs for LeCun's Initialization

Before presenting the proof of the convergence result under LeCun's initialization in Appendix C.3, let us state two helpful lemmas. The first lemma bounds the output of the network at initialization using standard Gaussian concentration and it is proved in Appendix C.1.

**Lemma C.1** *Let $\sigma$ be 1-Lipschitz, and consider LeCun's initialization scheme:*

$$[W_l]_{ij} \sim \mathcal{N}(0, 1/n_{l-1}), \quad \forall l \in [L], i \in [n_{l-1}], j \in [n_l].$$

*Fix some $t > 0$. Assume that $\sqrt{n_l} \geq t$ for any $l \in [L-1]$. Then,*

$$\|F_L\|_F \leq 2^{L-1} \frac{\|X\|_F}{\sqrt{d}} \left( \sqrt{n_L} + t \right), \tag{31}$$

*with probability at least $1 - Le^{-t^2/2}$.*

Recall the definition of $\lambda_F$:

$$\lambda_F = \sigma_{\min} \left( \sigma(XW_1^0) \right). \tag{32}$$

The second lemma identifies sufficient conditions on $n_1$ so that $\lambda_F$ is bounded away from zero. The proof is similar to that of Theorem 3.2 of [31] (see Section 6.8 in their appendix), and we provide it in Appendix C.2.

**Lemma C.2** *Let $|\sigma(x)| \leq |x|$ for every $x \in \mathbb{R}$. Define $F_1 = \sigma(XW)$ with $X \in \mathbb{R}^{N \times d}$, $W \in \mathbb{R}^{d \times n_1}$, and $W_{ij} \sim \mathcal{N}(0, \zeta^2)$ for all $i \in [d], j \in [n_1]$. Define also*

$$G_* = \mathbb{E}_{w \sim \mathcal{N}(0, \zeta^2 \mathbb{I}_d)} \left[ \sigma(Xw)\sigma(Xw)^T \right], \quad \lambda_* = \lambda_{min}(G_*).$$

*Then, for*

$$t \geq \sqrt{4\zeta^2 \ln \max \left( 1, 2\sqrt{6} \|X\|_2^2 d^{3/2} \zeta^2 \lambda_*^{-1} \right)}$$

*and*

$$n_1 \geq \max \left( N, \frac{20 \|X\|_2^2 dt^2 \left( t^2/2 + \ln(N/2) \right)}{\lambda_*} \right),$$

*we have*

$$\sigma_{min}(F_1) \geq \sqrt{n_1 \lambda_*/4} \tag{33}$$

*with probability at least $1 - 2e^{-t^2/2}$.*

### C.1 Proof of Lemma C.1

It is straightforward to show the following inequality.

**Lemma C.3** *Let $|\sigma(x)| \leq |x|$ for every $x \in \mathbb{R}$. Let $[W_l]_{ij} \sim \mathcal{N}\left( 0, \frac{1}{n_{l-1}} \right)$ for every $l \in [L], i \in [n_{l-1}], j \in [n_l]$. Then, for every $l \in [L]$ we have $\mathbb{E} \|F_l\|_F^2 \leq \frac{n_l}{n_{l-1}} \mathbb{E} \|F_{l-1}\|_F^2$.*

**Proof:**

$$\mathbb{E} \|F_l\|_F^2 = \mathbb{E} \|\sigma(F_{l-1}W_l)\|_F^2 \leq \mathbb{E} \|F_{l-1}W_l\|_F^2 = \mathbb{E} \operatorname{tr} \left( F_{l-1}W_l W_l^T F_{l-1}^T \right) = \frac{n_l}{n_{l-1}} \mathbb{E} \|F_{l-1}\|_F^2,$$

where the first inequality follows from our assumption on $\sigma$, and the last equality follows from the fact that $W_l W_l^T = \sum_{j=1}^{n_l} (W_l)_{:j}(W_l)_{:j}^T$ and $\mathbb{E}(W_l)_{:j}(W_l)_{:j}^T = \frac{1}{n_{l-1}} \mathbb{I}_{n_{l-1}}$ for every $j \in [n_l]$. $\quad\square$

**Proof of Lemma C.1.** In the following, we write $\operatorname{subG}(\xi^2)$ to denote a sub-gaussian random variable with mean zero and variance proxy $\xi^2$. It is well-known that if $Z \sim \operatorname{subG}(\xi^2)$ then for every $t \geq 0$ we have $\mathbb{P}(|Z| \geq t) \leq 2\exp(-\frac{t^2}{2\xi^2})$.

We prove by induction on $l \in [L]$ that, if $\sqrt{n_p} \geq t$ for every $p \in [l-1]$, then it holds w.p. $\geq 1 - le^{-t^2/2}$ over $(W_p)_{p=1}^l$ that

$$\|F_l\|_F \leq \frac{\|X_F\|}{\sqrt{d}} 2^{l-1} \left[ \sqrt{n_l} + t \right].$$

Let us check the case $l = 1$ first. We have

$$\left| \|F_1(W_1)\|_F - \|F_1(W_1')\|_F \right| \leq \|F_1(W_1) - F_1(W_1')\|_F$$
$$= \|\sigma(XW_1) - \sigma(XW_1')\|_F$$
$$\leq \|XW_1 - XW_1'\|_F \qquad \sigma \text{ is 1-Lipschitz}$$
$$\leq \|X\|_F \|W_1 - W_1'\|_F.$$

It follows that $\|F_1\|_F - \mathbb{E}\|F_1\|_F \sim \mathrm{subG}\left(\frac{\|X\|_F^2}{d}\right)$. By Gaussian concentration inequality, we have w.p. at least $1 - e^{-t^2/2}$,

$$
\begin{aligned}
\|F_1\|_F &\leq \mathbb{E}\|F_1\|_F + \frac{\|X\|_F}{\sqrt{d}}t \\
&\leq \frac{\sqrt{n_1}}{\sqrt{d}}\|X\|_F + \frac{\|X\|_F}{\sqrt{d}}t \qquad\qquad \text{Lemma C.3} \\
&= \frac{\|X\|_F}{\sqrt{d}}\left[\sqrt{n_1} + t\right].
\end{aligned}
$$

Thus the hypothesis holds for $l = 1$. Now suppose it holds for $l - 1$, that is, we have w.p. $\geq 1 - (l-1)e^{-t^2/2}$ over $(W_p)_{p=1}^{l-1}$,

$$
\|F_{l-1}\|_F \leq \frac{\|X\|_F}{\sqrt{d}}2^{l-2}\left[\sqrt{n_{l-1}} + t\right].
$$

Conditioned on $(W_p)_{p=1}^{l-1}$, we note that $\|F_l\|_F$ is Lipschitz w.r.t. $W_l$ because

$$
\left|\ \|F_l(W_l)\|_F - \|F_l(W_l')\|_F\ \right| \leq \|F_{l-1}\|_F\|W_l - W_l'\|_F
$$

and thus $\|F_l\|_F - \mathbb{E}\|F_l\|_F \sim \mathrm{subG}\left(\frac{\|F_{l-1}\|_F^2}{n_{l-1}}\right)$. By Gaussian concentration inequality, we have w.p. $\geq 1 - e^{-t^2/2}$ over $W_l$,

$$
\|F_l\|_F \leq \mathbb{E}\|F_l\|_F + \frac{\|F_{l-1}\|_F}{\sqrt{n_{l-1}}}t.
$$

Thus the above events hold w.p. at least $1 - le^{-t^2/2}$ over $(W_p)_{p=1}^{l}$, in which case we get

$$
\begin{aligned}
\|F_l\|_F &\leq \mathbb{E}\|F_l\|_F + \frac{\|F_{l-1}\|_F}{\sqrt{n_{l-1}}}t \\
&\leq \frac{\sqrt{n_l}}{\sqrt{n_{l-1}}}\|F_{l-1}\|_F + \frac{\|F_{l-1}\|_F}{\sqrt{n_{l-1}}}t \qquad\qquad \text{Lemma C.3} \\
&\leq \frac{\|X\|_F}{\sqrt{d}}2^{l-2}\left[\sqrt{n_{l-1}} + t\right]\frac{\sqrt{n_l} + t}{\sqrt{n_{l-1}}} \qquad\qquad \text{induction assump.} \\
&\leq \frac{\|X\|_F}{\sqrt{d}}2^{l-1}\left[\sqrt{n_l} + t\right] \qquad\qquad\qquad \sqrt{n_{l-1}} \geq t
\end{aligned}
$$

Thus, the hypothesis also holds for $l$. $\qquad\qquad\qquad\qquad\qquad\qquad\qquad\qquad\qquad$ $\square$

### C.2 Proof of Lemma C.2

Let $A \in \mathbb{R}^{N \times n_1}$ be a random matrix defined as $A_{:j} = \sigma(XW_{:j})\ \mathbb{1}_{\|W_{:j}\|_\infty \leq t}\ \forall j \in [n_1]$. Then,

$$
\lambda_{\min}\left(F_1 F_1^T\right) = \lambda_{\min}\left(\sum_{j=1}^{n_1}\sigma(XW_{:j})\sigma(XW_{:j})^T\right) \geq \lambda_{\min}\left(AA^T\right).
$$

Thus, by using our assumption on $\sigma$,

$$
\lambda_{\max}\left(A_{:j}A_{:j}^T\right) = \|A_{:j}\|_2^2 = \left\|\sigma(XW_{:j})\ \mathbb{1}_{\|W_{:j}\|_\infty \leq t}\right\|_2^2 \leq \|X\|_2^2\|W_{:j}\|_2^2\ \mathbb{1}_{\|W_{:j}\|_\infty \leq t} \leq \|X\|_2^2 dt^2 =: R.
$$

Let $G = \mathbb{E}_{w \sim \mathcal{N}(0, \zeta^2 \mathbb{I}_d)}\left[\sigma(Xw)\sigma(Xw)^T\ \mathbb{1}_{\|w\|_\infty \leq t}\right]$. Applying Matrix Chernoff bound (Theorem A.3) to the sum of random p.s.d. matrices, $AA^T = \sum_{j=1}^{n_1}A_{:j}A_{:j}^T$, we obtain that for every $\epsilon \in [0, 1)$

$$
\mathbb{P}\left(\lambda_{\min}\left(AA^T\right) \leq (1-\epsilon)\lambda_{\min}\left(\mathbb{E}AA^T\right)\right) \leq N\left[\frac{e^{-\epsilon}}{(1-\epsilon)^{1-\epsilon}}\right]^{\lambda_{\min}(\mathbb{E}AA^T)/R}.
$$

Substituting $\mathbb{E}[AA^T] = n_1 G$ and $R = \|X\|_2^2 \, dt^2$ and $\epsilon = 1/2$ gives

$$\mathbb{P}\Big(\lambda_{\min}\big(AA^T\big) \leq n_1 \lambda_{\min}(G)/2\Big) \leq N \left[\sqrt{2}e^{-1/2}\right]^{n_1 \lambda_{\min}(G)/R} \leq \exp\left(-\frac{n_1 \lambda_{\min}(G)}{10\|X\|_2^2 \, dt^2} + \ln N\right).$$

Thus, as long as $n_1$ is large enough, in particular,

$$n_1 \geq \frac{10\|X\|_2^2 \, dt^2 \Big(t^2/2 + \ln(N/2)\Big)}{\lambda_{\min}(G)},$$

we have $\lambda_{\min}\big(AA^T\big) \geq n_1 \lambda_{\min}(G)/2$ w.p. at least $1 - 2e^{-t^2/2}$.

The idea now is to lower bound $\lambda_{\min}(G)$ in terms of $\lambda_{\min}(G_*)$.

$$
\begin{aligned}
\|G - G_*\|_2 &= \left\|\mathbb{E}\left[\sigma(Xw)\sigma(Xw)^T \, \mathbb{1}_{\|w\|_\infty \leq t}\right] - \mathbb{E}\left[\sigma(Xw)\sigma(Xw)^T\right]\right\|_2 \\
&\leq \mathbb{E}\left\|\sigma(Xw)\sigma(Xw)^T \, \mathbb{1}_{\|w\|_\infty \leq t} - \sigma(Xw)\sigma(Xw)^T\right\|_2 && \text{Jensen inequality} \\
&= \mathbb{E}\left\|\sigma(Xw)\sigma(Xw)^T \, \mathbb{1}_{\|w\|_\infty > t}\right\|_2 \\
&= \mathbb{E}\left[\|\sigma(Xw)\|_2^2 \, \mathbb{1}_{\|w\|_\infty > t}\right] \\
&\leq \|X\|_2^2 \, \mathbb{E}\left[\|w\|_2^2 \, \mathbb{1}_{\|w\|_\infty > t}\right] && \text{assump. on } \sigma \\
&\leq \|X\|_2^2 \sqrt{\mathbb{E}[\|w\|_2^4] \, \mathbb{P}(\|w\|_\infty > t)} && \text{Cauchy-Schwarz} \\
&\leq \|X\|_2^2 \sqrt{d}\sqrt{\mathbb{E}\Big[\sum_{i=1}^{d} w_i^4\Big] \, \mathbb{P}(\|w\|_\infty > t)} && \text{Cauchy-Schwarz} \\
&= \|X\|_2^2 \, d\sqrt{3}\zeta^2 \sqrt{\mathbb{P}(\|w\|_\infty > t)} && \mathbb{E}_{x \sim \mathcal{N}(0,1)}[x^4] = 3 \\
&\leq \|X\|_2^2 \, d^{3/2}\zeta^2\sqrt{3}\sqrt{\mathbb{P}(|w_1| > t)} && \text{union bound} \\
&\leq \|X\|_2^2 \, d^{3/2}\zeta^2\sqrt{6}\exp\left(-\frac{t^2}{4\zeta^2}\right) && w_1 \sim \mathrm{subG}(\zeta^2) \\
&\leq \lambda_*/2 && \text{by assumpion on } t
\end{aligned}
$$

This implies that $\lambda_{\min}(G) \geq \lambda_{\min}(G_*) - \lambda_*/2 = \lambda_*/2$. Plugging this into the above statement yields for every

$$n_1 \geq \frac{20\|X\|_2^2 \, dt^2 \Big(t^2/2 + \ln(N/2)\Big)}{\lambda_*},$$

it holds w.p. at least $1 - 2e^{-t^2/2}$ that

$$
\begin{aligned}
\lambda_{\min}\big(F_1 F_1^T\big) &\geq \lambda_{\min}\big(AA^T\big) \\
&\geq n_1 \lambda_{\min}(G)/2 \\
&\geq n_1(\lambda_{\min}(G_*) - \lambda_*/2)/2 \\
&\geq n_1 \lambda_*/4.
\end{aligned}
$$

Lastly, since $n_1 \geq N$ we get $\sigma_{\min}(F_1) = \sqrt{\lambda_{\min}\big(F_1 F_1^T\big)} \geq \sqrt{n_1 \lambda_*/4}$. $\qquad\square$

## C.3 Formal statement and proof for LeCun's Initialization

**Theorem C.4** *Let the activation function satisfy Assumption 2.2. Fix $t > 0$, $t_0 \geq \max\left\{1, \sqrt{4d^{-1}\ln\max\left(1, 2\sqrt{6d}\|X\|_2^2 \lambda_*^{-1}\right)}\right\}$, and denote by $c$ a large enough constant depending only on the parameters $\gamma, \beta$ of the activation function. Let the widths of the neural network satisfy*

*the following conditions:*

$$\sqrt{n_{l-1}} \geq \left(1 + \frac{1}{100}\right)(\sqrt{n_l} + t), \quad \forall l \in \{2, \dots, L\}, \tag{34}$$

$$n_1 \geq \max\left(N, \, d, \, \frac{ct_0^2 d \|X\|_2^2 \left(t_0^2 + \ln N\right)}{\lambda_*}, \, \frac{2^{cL} \|X\|_F^2}{d\lambda_*^2} \left(\frac{(\sqrt{n_L} + t) \|X\|_F}{\sqrt{d}} + \|Y\|_F\right)^2\right). \tag{35}$$

*Let us consider LeCun's initialization:*

$$[W_l^0]_{ij} \sim \mathcal{N}(0, 1/n_{l-1}), \quad \forall l \in [L], i \in [n_{l-1}], j \in [n_l].$$

*Let the learning rate satisfy*

$$\eta < \left(\frac{2^{cL} n_1}{d} \cdot \max(1, \|X\|_F^2) \cdot \max\left(1, \frac{(\sqrt{n_L} + t) \|X\|_F}{\sqrt{d}}, \|Y\|_F\right)\right)^{-1}. \tag{36}$$

*Then, the training loss vanishes and the network parameters converge to a global minimizer $\theta_*$ at a geometric rate as*

$$\Phi(\theta_k) \leq \left(1 - \frac{\eta n_1 \lambda_*}{2^{cL}}\right)^k \Phi(\theta_0), \tag{37}$$

$$\|\theta_k - \theta_*\|_2 \leq \left(1 - \frac{\eta n_1 \lambda_*}{2^{cL}}\right)^{k/2} 2^{cL} \frac{\|X\|_F}{\sqrt{n_1 d}\lambda_*} \cdot \left(\frac{(\sqrt{n_L} + t) \|X\|_F}{\sqrt{d}} + \|Y\|_F\right), \tag{38}$$

*with probability at least $1 - 3Le^{-t^2/2} - 2e^{-t_0^2/2}$.*

Before presenting the proof of Theorem C.4, let us explain how to derive (12) from the main paper.

**How to derive** (12) **from Theorem C.4.** For the convenience of the reader, we recall that in the discussion of Section 3.2 from the main paper, in order to get (12), the following standard setting has been considered: *(i)* $N \geq d$, *(ii)* the training samples lie on the sphere of radius $\sqrt{d}$, *(iii)* $n_L$ is a constant, and *(iv)* the target labels satisfy $\|y_i\| = \mathcal{O}(1)$ for all $i \in [N]$. It follows from *(i)* and *(ii)* that $\|X\|_2^2 \leq \|X\|_F^2 = Nd \leq N^2$. Thus we have that

$$\sqrt{4d^{-1} \ln \max\left(1, 2\sqrt{6d} \|X\|_2^2 \lambda_*^{-1}\right)} = \mathcal{O}\left(\sqrt{d^{-1} \ln(N\lambda_*^{-1})}\right). \tag{39}$$

This implies that

$$\frac{ct_0^2 d \|X\|_2^2 \left(t_0^2 + \ln N\right)}{\lambda_*} = \mathcal{O}\left(\frac{\|X\|_2^2}{\lambda_*} \left(\log \frac{N}{\lambda_*}\right)^2\right). \tag{40}$$

Furthermore, from *(iii)* and *(iv)* we have that

$$\frac{2^{cL} \|X\|_F^2}{d\lambda_*^2} \left(\frac{(\sqrt{n_L} + t) \|X\|_F}{\sqrt{d}} + \|Y\|_F\right)^2 = \mathcal{O}\left(\frac{N^2 2^{\mathcal{O}(L)}}{\lambda_*^2}\right). \tag{41}$$

By combining (40) and (41), the scaling (12) follows from the condition (35).

**Proof of Theorem C.4.** From known results on random Gaussian matrices, we have, w.p. $\geq 1 - 2e^{-t^2/2}$,

$$\|W_1^0\|_2 \leq \frac{\sqrt{n_1} + \sqrt{d} + t}{\sqrt{d}} \leq 3\frac{\sqrt{n_1}}{\sqrt{d}},$$

$$\|W_2^0\|_2 \leq \frac{\sqrt{n_1} + \sqrt{n_2} + t}{\sqrt{n_1}} \leq 2,$$

where the last inequality in each line follows from $n_1 \geq d$ and from (34). From def. (3), we get

$$\bar{\lambda}_1 = \frac{2}{3}(1 + \|W_1^0\|_2) \leq \frac{8}{3}\frac{\sqrt{n_1}}{\sqrt{d}}, \tag{42}$$

$$\bar{\lambda}_2 = \frac{2}{3}(1 + \|W_2^0\|_2) \leq 2.$$

Similarly, for any $l \in \{3, \ldots, L\}$, we have, w.p. $\geq 1 - 2e^{-t^2/2}$,

$$\frac{1}{101} \leq \frac{\sqrt{n_{l-1}} - \sqrt{n_l} - t}{\sqrt{n_{l-1}}} \leq \lambda_l \leq \bar{\lambda}_l \leq \frac{\sqrt{n_{l-1}} + \sqrt{n_l} + t}{\sqrt{n_{l-1}}} \leq 2. \tag{43}$$

Furthermore, by Lemma C.1 and C.2, we have, w.p. $\geq 1 - Le^{-t^2/2} - 2e^{-t_0^2/2}$,

$$\lambda_F = \sigma_{\min}\left(F_1^0\right) \geq \sqrt{n_1 \lambda_*/4}, \tag{44}$$

$$\sqrt{2\Phi(\theta_0)} \leq 2^{L-1}(\sqrt{n_L} + t)\frac{\|X\|_F}{\sqrt{d}} + \|Y\|_F, \tag{45}$$

as long as the width of the first layer satisfies the following condition from Lemma C.2:

$$n_1 \geq \max\left(N, \frac{ct_0^2 d \|X\|_2^2 \left(t_0^2 + \ln N\right)}{\lambda_*}\right), \tag{46}$$

for a suitable constant $c$. From (44), we get a lower bound on the LHS of (4); and from (42), (43) and (45) we get an upper bound on the RHS of (4). Thus in order to satisfy the initial condition (4), it suffices to have (46) and

$$n_1 \lambda_* \geq 2^{cL} \|X\|_F \sqrt{\frac{n_1}{d}} \left((\sqrt{n_L} + t)\frac{\|X\|_F}{\sqrt{d}} + \|Y\|_F\right), \tag{47}$$

which together leads to condition (35).

To satisfy the initial condition (5), it suffices to have in addition to (4) that $\lambda_F \geq 2\|X\|_2$, which is fulfilled for $n_1 \geq \frac{16\|X\|_2^2}{\lambda_*}$, which is however satisfied by (35) already.

As a result, the initial conditions (4)-(5) are satisfied and we can apply Theorem 3.2. Let us now bound the quantities $\alpha_0, Q_0$ and $Q_1$ defined in (6). Note that $\lambda_F = \sigma_{\min}\left(\sigma(XW_1^0)\right) \leq \left\|\sigma(XW_1^0)\right\|_F \leq \|X\|_F \|W_1^0\|_2$. Then,

$$\frac{n_1 \lambda_*}{2^{cL}} \leq \alpha_0 \leq 2^{cL} \|X\|_F^2 \frac{n_1}{d}, \tag{48}$$

and

$$Q_0 \leq 2^{cL} \|X\|_F^2 \frac{n_1}{d} + 2^{cL}\frac{n_1}{d} \|X\|_F (1 + \|X\|_F)\sqrt{2\Phi(\theta_0)}$$

$$\leq \frac{2^{cL} n_1}{d} \max(1, \|X\|_F^2) \max\left(1, \frac{(\sqrt{n_L} + t)\|X\|_F}{\sqrt{d}}, \|Y\|_F\right) \qquad \text{by (45)}.$$

It is easy to see that the upper bound of $Q_0$ dominates that of $\alpha_0$. Thus to satisfy the learning rate condition from Theorem 3.2, it suffices to set $\eta$ to be smaller than the inverse of the upper bound on $Q_0$, which leads to condition (36).

From the lower bound of $\alpha_0$ in (48) and (7), we immediately get the convergence of the loss as stated in (37). Similarly, one can compute the quantity $Q_1$ defined in (8) to get the convergence of the parameters as stated in (38). $\qquad \square$

# D   Proofs for Lower Bound on $\lambda_*$

## D.1   Background on Hermite Expansions

Let $L^2(\mathbb{R}, w(x))$ denote the set of all functions $f : \mathbb{R} \to \mathbb{R}$ such that

$$\int_{-\infty}^{\infty} f^2(x)w(x)dx < \infty.$$

The normalized probabilist's hermite polynomials are given by

$$h_r(x) = \frac{1}{\sqrt{r!}}(-1)^r e^{x^2/2} \frac{d^r}{dx^r} e^{-x^2/2}.$$

The functions $\{h_r(x)\}_{r=0}^{\infty}$ form an orthonormal basis of $L^2\left(\mathbb{R}, \frac{e^{-x^2/2}}{\sqrt{2\pi}}\right)$, which is a Hilbert space with the inner product

$$\langle \sigma_1, \sigma_2 \rangle = \int_{-\infty}^{\infty} \sigma_1(x)\sigma_2(x) \frac{e^{-x^2/2}}{\sqrt{2\pi}} dx.$$

Thus, every function $\sigma$ in $L^2\left(\mathbb{R}, \frac{e^{-x^2/2}}{\sqrt{2\pi}}\right)$ can be represented as (a.k.a. Hermite expansion):

$$\sigma(x) = \sum_{r=0}^{\infty} \mu_r(\sigma)h_r(x), \tag{49}$$

where $\mu_r(\sigma)$ is the $r$-th Hermite coefficient given by

$$\mu_r(\sigma) = \int_{-\infty}^{\infty} \sigma(y)h_r(y) \frac{e^{-y^2/2}}{\sqrt{2\pi}} dy.$$

Let $\|\cdot\|$ be defined as $\|\sigma\|^2 = \langle \sigma, \sigma \rangle$. Then, the convergence of the series in (49) is understood in the sense that

$$\lim_{n \to \infty} \left\| \sigma(x) - \sum_{r=0}^{n} \mu_r(\sigma)h_r(x) \right\| = \lim_{n \to \infty} \mathbb{E}_{x \sim \mathcal{N}(0,1)} \left| \sigma(x) - \sum_{r=0}^{n} \mu_r(\sigma)h_r(x) \right|^2 = 0$$

Note $\sigma \in L^2\left(\mathbb{R}, \frac{e^{-x^2/2}}{\sqrt{2\pi}}\right)$ if and only if $\langle \sigma, \sigma \rangle = \sum_{r=0}^{\infty} \mu_r^2(\sigma) < \infty$.

**Lemma D.1** *Consider a Hilbert space $H$ equipped with an inner product $\langle \cdot, \cdot \rangle : H \times H \to \mathbb{R}$. Let $\|\cdot\|$ be norm induced by the inner product, i.e. $\|f\| = \sqrt{\langle f, f \rangle}$. Let $\{f_n\}, \{g_n\}$ be two sequences in $H$ such that $\lim_{n \to \infty} \|f_n - f\| = \lim_{n \to \infty} \|g_n - g\| = 0$. Then $\langle f, g \rangle = \lim_{n \to \infty} \langle f_n, g_n \rangle$.*

**Proof:**

$$
\begin{aligned}
|\langle f, g \rangle - \langle f_n, g_n \rangle| &\leq |\langle f, g - g_n \rangle| + |\langle f - f_n, g_n \rangle| \\
&\leq \|f\| \|g - g_n\| + \|f - f_n\| \|g_n\| \\
&\leq \|f\| \|g - g_n\| + \|f - f_n\| (\|g_n - g\| + \|g\|).
\end{aligned}
$$

Taking the limit on both sides yields the result. $\qquad\square$

**Lemma D.2** *Let $x, y \in \mathbb{R}^d$ be such that $\|x\|_2 = \|y\|_2 = 1$. Then, for every $j, k \geq 0$,*

$$\mathbb{E}_{w \sim \mathcal{N}(0, \mathbb{I}_d)}\left[h_j(\langle w, x \rangle)h_k(\langle w, y \rangle)\right] = \begin{cases} \langle x, y \rangle^j & j = k \\ 0 & j \neq k \end{cases}.$$

**Proof:** Let $s, t \in \mathbb{R}$ be given finite variables. Then,

$$
\begin{aligned}
\mathbb{E} \exp\left(s \langle w, x \rangle + t \langle w, y \rangle\right) &= \prod_{i=1}^{d} \mathbb{E} \exp\left(w_i(sx_i + ty_i)\right) \\
&= \prod_{i=1}^{d} \exp\left(\frac{(sx_i + ty_i)^2}{2}\right) \\
&= \exp\left(\frac{s^2 + t^2 + 2st \langle x, y \rangle}{2}\right).
\end{aligned}
$$

Thus, it follows that

$$\mathbb{E}\exp\left(s\langle w,x\rangle-\frac{s^2}{2}\right)\exp\left(t\langle w,y\rangle-\frac{t^2}{2}\right)=\exp\left(st\langle x,y\rangle\right). \tag{50}$$

Let $L^2(\mathbb{R}^d)$ be the space of functions $f:\mathbb{R}^d\to\mathbb{R}$ with bounded gaussian measure, i.e.

$$\mathbb{E}_{w\sim\mathcal{N}(0,\mathbb{I}_d)}[f(w)^2]<\infty.$$

This is a Hilbert space w.r.t. the inner product $\langle f,g\rangle=\mathbb{E}[fg]$ and its induced norm $\|f\|=\sqrt{\langle f,f\rangle}$. Let the functions $f,g:\mathbb{R}^d\to\mathbb{R}$ be defined as

$$f(w)=\exp\left(s\langle w,x\rangle-\frac{s^2}{2}\right),\quad g(w)=\exp\left(t\langle w,y\rangle-\frac{t^2}{2}\right).$$

Then the LHS of (50) becomes $\langle f,g\rangle$. Let $\{f_n\}_{n=1}^\infty,\{g_n\}_{n=1}^\infty$ be two sequence of functions defined as

$$f_n(w)=\sum_{j=0}^n h_j(\langle w,x\rangle)\frac{s^j}{\sqrt{j!}},\quad g_n(w)=\sum_{k=0}^n h_k(\langle w,y\rangle)\frac{t^k}{\sqrt{k!}}.$$

One can easily check that $f,g$ are in $L^2(\mathbb{R}^d)$, and so are $f_n$'s and $g_n$'s. Moreover,

$$\lim_{n\to\infty}\|f_n-f\|^2=\lim_{n\to\infty}\mathbb{E}_{w\sim\mathcal{N}(0,\mathbb{I}_d)}|f_n(w)-f(w)|^2$$

$$=\lim_{n\to\infty}\mathbb{E}_{u\sim\mathcal{N}(0,1)}\left|\exp\left(su-\frac{s^2}{2}\right)-\sum_{j=0}^n h_j(u)\frac{s^j}{\sqrt{j!}}\right|^2$$

$$=0,$$

where the last equality follows from the Hermite expansion of the function $u\mapsto\exp(su-s^2/2)$, which is given by

$$\exp\left(su-\frac{s^2}{2}\right)=\sum_{j=0}^\infty h_j(u)\frac{s^j}{\sqrt{j!}}.$$

Similarly, $\lim_{n\to\infty}\|g_n-g\|^2=0$. By applying Lemma D.1 and taking the Mclaurin series of the RHS of (50), we obtain

$$\sum_{j,k=0}^\infty\mathbb{E}\frac{h_j(\langle w,x\rangle)h_k(\langle w,y\rangle)}{\sqrt{j!k!}}s^jt^k=\sum_{j=0}^\infty\frac{\langle x,y\rangle^j}{j!}s^jt^j,\quad\forall s,t\in\mathbb{R}.$$

Equating the coefficients on both sides gives the desired result. $\qquad\square$

### D.2   Formal statement and proof of (16)

**Lemma D.3** *Let $X=[x_1,\ldots,x_N]^T\in\mathbb{R}^{N\times d}$ where $\|x_i\|_2=\sqrt{d}$ for all $i\in[N]$. Assume that $\sigma\in L^2(\mathbb{R},e^{-x^2/2}/\sqrt{2\pi})$. Let $G_*$ be defined as in (13). Then,*

$$G_*=\sum_{r=0}^\infty\frac{\mu_r^2(\sigma)}{d^r}(X^{*r})(X^{*r})^T.$$

*Here, "=" is understood in the sense of uniform convergence, that is, for every $\epsilon>0$, there exists a sufficiently large $r_0\geq 0$ such that*

$$\left|(G_*)_{ij}-(S_r)_{ij}\right|<\epsilon,\quad\forall i,j\in[N],\forall r\geq r_0,$$

*where $S_r=\sum_{k=0}^r\frac{\mu_k^2(\sigma)}{d^k}(X^{*k})(X^{*k})^T$.*

This result is also stated in Lemma H.2 of [31] for ReLU and softplus activation functions. As a fully rigorous proof is missing in [31], we provide it below.

**Proof of Lemma D.3.** Let $\bar{x}_i = x_i / \|x_i\|_2$ for $i \in [N]$. From the definition of $G_*$, we have

$$(G_*)_{ij} = \mathbb{E}_{w \sim \mathcal{N}(0, \mathbb{I}_d / d)} \left[ \sigma(\langle w, x_i \rangle) \sigma(\langle w, x_j \rangle) \right]$$

$$= \mathbb{E}_{\bar{w} \sim \mathcal{N}(0, \mathbb{I}_d)} \left[ \sigma(\langle \bar{w}, \bar{x}_i \rangle) \sigma(\langle \bar{w}, \bar{x}_j \rangle) \right] \qquad \bar{w} = \sqrt{d} w$$

$$= \sum_{r,s=0}^{\infty} \mu_r(\sigma) \mu_s(\sigma) \mathbb{E}_{\bar{w} \sim \mathcal{N}(0, \mathbb{I}_d)} \left[ h_r(\langle \bar{w}, \bar{x}_i \rangle) h_s(\langle \bar{w}, \bar{x}_j \rangle) \right] \qquad (*)$$

$$= \sum_{r=0}^{\infty} \mu_r^2(\sigma) \ \langle \bar{x}_i, \bar{x}_j \rangle^r \qquad \text{Lemma D.2}$$

where $(*)$ is justified below. Note that $\langle \bar{x}_i, \bar{x}_j \rangle^r = \frac{1}{d^r} \langle x_i \otimes \ldots \otimes x_i, x_j \otimes \ldots \otimes x_j \rangle$. Thus,

$$G_* = \sum_{r=0}^{\infty} \frac{\mu_r^2(\sigma)}{d^r} (X^{*r})(X^{*r})^T.$$

To justify step $(*)$, we can use the similar argument as in the proof of Lemma D.2. Indeed, consider the same Hilbert space $L^2(\mathbb{R}^d)$ as defined there. Let $f, g : \mathbb{R}^d \to \mathbb{R}$ be defined as

$$f(\bar{w}) = \sigma(\langle \bar{w}, \bar{x}_i \rangle), \quad g(\bar{w}) = \sigma(\langle \bar{w}, \bar{x}_j \rangle).$$

and the sequence $\{f_n\}, \{g_n\}$ defined as

$$f_n(\bar{w}) = \sum_{r=0}^{n} \mu_r(\sigma) h_r(\langle \bar{w}, \bar{x}_i \rangle), \quad g_n(\bar{w}) = \sum_{s=0}^{n} \mu_s(\sigma) h_s(\langle \bar{w}, \bar{x}_j \rangle).$$

It is easy to see that $f, g, \{f_n\}, \{g_n\} \in L^2(\mathbb{R}^d)$. Moreover, $\|f_n - f\|^2 = \mathbb{E}_{z \sim \mathcal{N}(0,1)} |\sigma(z) - \sum_{r=0}^{n} \mu_r(\sigma) h_r(z)|^2 \to 0$ as $n \to \infty$. Similarly, $\|g_n - g\|^2 \to 0$ as $n \to \infty$. Thus applying Lemma D.1 leads us to $(*)$.

$\square$

## D.3 Proof of Lemma 3.4

Define $K = X^{*r}$ and note that, for $i \in [N]$, the $i$-th row of $K$ is given by the $r$-th Kronecker power of $x_i$, namely, $K_{i:} = x_i^{\otimes r} = x_i \otimes x_i \otimes \cdots \otimes x_i \in \mathbb{R}^{d^r}$. Let $z = (z_1, \ldots, z_N) \in \mathbb{R}^N$ be such that $\|z\|_2 = 1$. Then,

$$\|K^T z\|_2^2 = \sum_{i=1}^{N} z_i^2 \|K_{i:}\|_2^2 + \sum_{i \neq j} \langle z_i K_{i:}, z_j K_{j:} \rangle$$

$$= \sum_{i=1}^{N} z_i^2 \|x_i\|_2^{2r} + \sum_{i \neq j} z_i z_j \langle x_i, x_j \rangle^r \qquad (51)$$

$$= d^r + \sum_{i \neq j} z_i z_j \langle x_i, x_j \rangle^r.$$

Furthermore, we have that

$$\left| \sum_{i \neq j} z_i z_j \langle x_i, x_j \rangle^r \right| \leq \sum_{i \neq j} |z_i| |z_j| |\langle x_i, x_j \rangle|^r$$

$$\leq (\max_{i \neq j} |\langle x_i, x_j \rangle|)^r \left( \sum_{i=1}^{N} |z_i| \right)^2 \qquad (52)$$

$$\leq N (\max_{i \neq j} |\langle x_i, x_j \rangle|)^r,$$

where in the last step we have used Cauchy-Schwarz inequality and that $\|z\|_2 = 1$. By combining (51) and (52), we obtain that

$$\sigma_{\min}^2(K) \geq d^r - N \left( \max_{i \neq j} |\langle x_i, x_j \rangle| \right)^r. \tag{53}$$

Let us now provide a bound on $\max_{i \neq j} |\langle x_i, x_j \rangle|$. Fix any $u \in \mathbb{R}^d$ such that $\|u\|_2 = \sqrt{d}$, and recall that, by hypothesis, $\|x_i\|_{\psi_2} \leq c_1$, where $c_1$ is a constant that does not depend on $d$. Then, for all $t \geq 0$,

$$\mathbb{P}\left( |\langle x_i, u \rangle| \geq t\sqrt{d} \right) \leq 2e^{-C_1 t^2}, \tag{54}$$

where $C_1$ is a constant that does not depend on $d$. As $x_i$ and $x_j$ are independent for $i \neq j$ and $\|x_j\|_2 = \sqrt{d}$, we deduce that

$$\mathbb{P}\left( |\langle x_i, x_j \rangle| \geq t\sqrt{d} \right) \leq 2e^{-C_1 t^2}. \tag{55}$$

By doing a union bound, we have that

$$\mathbb{P}\left( \max_{i \neq j} |\langle x_i, x_j \rangle| \geq t\sqrt{d} \right) \leq 2N^2 e^{-C_1 t^2}, \tag{56}$$

which, combined with (53), yields

$$\mathbb{P}\left( \sigma_{\min}^2(K) \geq d^r - Nt^r d^{r/2} \right) \leq 2N^2 e^{-C_1 t^2}. \tag{57}$$

Thus, by taking $t = \left( \frac{3d^{r/2}}{4N} \right)^{1/r}$, the proof is complete. $\qquad \square$

## D.4 Proof of Theorem 3.3

First, we show that, if $\sigma$ is not a linear function and $|\sigma(x)| \leq |x|$ for $x \in \mathbb{R}$, then $\mu_r(\sigma) > 0$ for arbitrarily large $r$.

**Lemma D.4** *Assume that $\sigma$ is not a linear function, and that $|\sigma(x)| \leq |x|$ for every $x \in \mathbb{R}$. Then,*

$$\sup \{ r \mid \mu_r(\sigma) > 0 \} = \infty. \tag{58}$$

**Proof:** It suffices to show that $\sigma$ cannot be represented by any polynomial of finite degree. Suppose, by contradiction, that $\sigma(x) = \sum_{i=0}^{n} a_i x^i$, where $a_n \neq 0$ and $n \geq 2$. As $\sigma(0) = 0$, we have that $a_0 = 0$. Thus,

$$\begin{aligned}
\lim_{x \to \infty} \frac{|\sigma(x)|}{|x|} &= \lim_{x \to \infty} \left| a_n x^{n-1} + \ldots + a_1 \right| \\
&= \lim_{x \to \infty} |x|^{n-1} \frac{\left| a_n x^{n-1} + \ldots + a_1 \right|}{|x|^{n-1}} \\
&= \lim_{x \to \infty} |x|^{n-1} \left| a_n + \ldots + \frac{a_1}{x^{n-1}} \right| \\
&= \infty.
\end{aligned}$$

This contradicts the fact that $\frac{|\sigma(x)|}{|x|}$ is bounded, and it concludes the proof. $\qquad \square$

At this point, we are ready to prove Theorem 3.3.

**Proof of Theorem 3.3.** As $\sigma$ is not linear and $|\sigma(x)| \leq |x|$ for every $x \in \mathbb{R}$, by Lemma D.4, we have that $\sup \{ r \mid \mu_r(\sigma) > 0 \} = \infty$. Thus, there there exists an integer $r \geq 10k$ such that $\mu_r(\sigma) \neq 0$. Thus, Lemma 3.4 implies that, for $N \leq d^r$,

$$\lambda_{\min}\left( (X^{*r})(X^{*r})^T \right) = \sigma_{\min}^2 (X^{*r}) \geq d^r / 4,$$

with probability at least

$$1 - 2N^2 e^{-c_2 d N^{-2/r}} \geq 1 - 2N^2 e^{-c_2 N^{4/5k}},$$

where in the last step we use that $N \leq d^k$ and $r \geq 10k$.

Furthermore, by Lemma D.3, there exists $r' \geq r$ such that

$$\|G_* - S_{r'}\|_F < \frac{\mu_r^2(\sigma)}{2d^r} \lambda_{\min}\left((X^{*r})(X^{*r})^T\right) =: \frac{\xi}{2}.$$

Note that $\lambda_{\min}(S_{r'}) \geq \lambda_{\min}(S_r) \geq \xi$. Thus by Weyl's inequality, we get

$$\lambda_* = \lambda_{\min}(G_*) \geq \lambda_{\min}(S_{r'}) - \frac{\xi}{2} \geq \frac{\xi}{2} \geq \frac{\mu_r^2(\sigma)}{8},$$

which completes the proof. $\qquad\qquad\qquad\qquad\qquad\qquad\qquad\qquad\qquad\qquad\qquad\qquad\square$

### D.5 Improvement of Lemma 3.4 for $r \in \{2, 3, 4\}$

The goal of this section is to prove the following result.

**Lemma D.5** *Let $X \in \mathbb{R}^{N \times d}$ be a matrix whose rows are i.i.d. random vectors uniformly distributed on the sphere of radius $\sqrt{d}$. Fix an integer $r \geq 2$. Then, there exists $c_1 \in (0, 1)$ such that, for $d \leq N \leq c_1 d^2$, we have*

$$\mathbb{P}\left(\sigma_{\min}(X^{*r}) \geq d^{r/2}/2\right) \geq 1 - 2Ne^{-c_2 d^{1/r}} - (1 + 3\log N)e^{-11\sqrt{N}} \qquad (59)$$

*for some constant $c_2 > 0$.*

Let us emphasize that the constants $c_1, c_2 > 0$ do not depend on $N$ and $d$, but they can depend on the integer $r$. Note that the probability in the RHS of (59) tends to 1 as long as $N$ is $\mathcal{O}(d^2)$. Thus, this result improves upon Lemma 3.4 for $r \in \{2, 3, 4\}$. The price to pay is a stronger assumption on $X$. In fact, Lemma D.5 requires that the rows of $X$ are uniformly distributed on the sphere of radius $\sqrt{d}$, while Lemma 3.4 only requires that they are sub-Gaussian. Recall that the sub-Gaussian norm of a vector uniformly distributed on the sphere of radius $\sqrt{d}$ is a constant (independent of $d$), see Theorem 3.4.6 in [42]. Thus, the requirement on $X$ of Lemma D.5 is strictly stronger than that of Lemma 3.4.

Recall that, given a random variable $Y \in \mathbb{R}$, its sub-exponential norm is defined as

$$\|Y\|_{\psi_1} = \inf\{C > 0 : \mathbb{E}[e^{|Y|/C}] \leq 2\}. \qquad (60)$$

Furthermore, for a centered random vector $x \in \mathbb{R}^d$, its sub-exponential norm is defined as

$$\|x\|_{\psi_1} = \sup_{\|y\|_2=1} \|\langle x, y \rangle\|_{\psi_1}. \qquad (61)$$

We start by stating two intermediate results that will be useful for the proof.

**Lemma D.6** *Consider an $r$-indexed matrix $A = (a_{i_1,\dots,i_r})_{i_1,\dots,i_r=1}^d$ such that $a_{i_1,\dots,i_r} = 0$ whenever $i_j = i_k$ for some $j \neq k$. Let $x = (x_1, \dots, x_d)$ be a random vector in $\mathbb{R}^d$ uniformly distributed on the unit sphere, and define*

$$Z = \sum_{\boldsymbol{i} \in [d]^r} a_{\boldsymbol{i}} \prod_{j=1}^r x_{i_j}. \qquad (62)$$

*Then,*

$$\mathbb{E}\left[e^{Cd|Z|^{2/r}}\right] \leq 2, \qquad (63)$$

*where $C$ is a numerical constant.*

If $x$ is uniformly distributed on the unit sphere, then it satisfies the logarithmic Sobolev inequality with constant $2/d$, see Corollary 1.1 in [15]. Thus, Lemma D.6 follows from Theorem 1.14 in [9], where $\sigma^2 = 1/d$ (see (1.18) in [9]) and the function $f$ is a homogeneous tetrahedral polynomial of degree $r$.

The second intermediate lemma is stated below and it follows from Theorem 5.1 of [1] (this is also basically a restatement of Lemma F.2 of [37]).

**Lemma D.7** *Let* $u_1, u_2, \ldots, u_N$ *be independent sub-exponential random vectors with* $\psi = \max_{i \in [N]} \|u_i\|_{\psi_1}$. *Let* $\eta_{\max} = \max_{i \in [N]} \|u_i\|_2$ *and define*

$$B_N = \sup_{z\,:\,\|z\|_2 = 1} \left| \sum_{i \neq j} \langle z_i u_i, z_j u_j \rangle \right|^{1/2}. \tag{64}$$

*Then,*

$$\mathbb{P}\left( B_N^2 \geq \max(B^2, \eta_{\max} B, \eta_{\max}^2/4) \right) \leq (1 + 3\log N)e^{-11\sqrt{N}}, \tag{65}$$

*where*

$$B = C_0 \psi \sqrt{N}, \tag{66}$$

*and* $C_0$ *is a numerical constant.*

At this point, we are ready to provide the proof of Lemma D.5.

**Proof of Lemma D.5.** The first step is to drop columns from $X^{*r}$. Define $K = X^{*r}$ and note that, for $i \in [N]$, the $i$-th row of $K$ is given by the $r$-th Kronecker power of $x_i$, namely, $x_i^{\otimes r} = x_i \otimes x_i \otimes \cdots \otimes x_i \in \mathbb{R}^{d^r}$. Let $x_i = (x_{i,1}, \ldots, x_{i,d})$ and index the columns of $K$ as $(j_1, j_2, \ldots, j_r)$, with $j_p \in [d]$ for all $p \in [r]$, so that the element of $K$ in row $i$ and column $(j_1, j_2, \ldots, j_r)$ is given by $\prod_{p=1}^r x_{i,j_p}$. Consider the matrix $\tilde{K}$ obtained by keeping only the columns of $K$ where the indices $j_1, j_2, \ldots j_r$ are all different. Note that $\tilde{K}$ has $\prod_{j=0}^{r-1}(d - j) \geq N$ columns as $N \leq c_1 d^2$. Thus, as $\tilde{K} = X^{*r}$ is obtained by dropping columns from $K$, then

$$\sigma_{\min}(K) \geq \sigma_{\min}(\tilde{K}). \tag{67}$$

The second step is to bound the sub-exponential norm of the rows of $\tilde{K}$. Let $\tilde{k}_x$ be the row of $\tilde{K}$ corresponding to the data point $x = (x_1, \ldots, x_d)$. Let us emphasize that, from now till the end of the proof, we denote by $x_i \in \mathbb{R}$ the $i$-th element of the vector $x$ (and not the $i$-th training sample, which is a vector in $\mathbb{R}^d$). Let $\mathcal{A}$ be the set of $r$-indexed matrices $A = (a_{i_1, \ldots, i_r})_{i_1, \ldots, i_r = 1}^d$ such that $\sum_{\boldsymbol{i} \in [d]^r} a_{\boldsymbol{i}}^2 = 1$ and $a_{i_1, \ldots, i_r} = 0$ whenever $i_j = i_k$ for some $j \neq k$. Then, by definition of sub-exponential norm of a vector, we have that

$$\|\tilde{k}_x\|_{\psi_1} = \sup_{A \in \mathcal{A}} \left\| \sum_{\boldsymbol{i} \in [d]^r} a_{\boldsymbol{i}} \prod_{j=1}^r x_{i_j} \right\|_{\psi_1}. \tag{68}$$

Note that, for all $A \in \mathcal{A}$,

$$\left| \sum_{\boldsymbol{i} \in [d]^r} a_{\boldsymbol{i}} \prod_{j=1}^r x_{i_j} \right| \overset{(a)}{\leq} \sqrt{\sum_{\boldsymbol{i} \in [d]^r} a_{\boldsymbol{i}}^2} \sqrt{\sum_{\boldsymbol{i} \in [d]^r} \prod_{j=1}^r x_{i_j}^2} \overset{(b)}{=} d^{r/2}, \tag{69}$$

where in (a) we use Cauchy-Schwarz inequality and in (b) we use that $\sum_{\boldsymbol{i} \in [d]^r} a_{\boldsymbol{i}}^2 = 1$ and $\|x\|_2 = \sqrt{d}$. Consequently,

$$\|\tilde{k}_x\|_{\psi_1} = \sup_{A \in \mathcal{A}} \left\| \left| \sum_{\boldsymbol{i} \in [d]^r} a_{\boldsymbol{i}} \prod_{j=1}^r x_{i_j} \right|^{1-2/r} \left| \sum_{\boldsymbol{i} \in [d]^r} a_{\boldsymbol{i}} \prod_{j=1}^r x_{i_j} \right|^{2/r} \right\|_{\psi_1}$$
$$\leq d^{r/2 - 1} \sup_{A \in \mathcal{A}} \left\| \left| \sum_{\boldsymbol{i} \in [d]^r} a_{\boldsymbol{i}} \prod_{j=1}^r x_{i_j} \right|^{2/r} \right\|_{\psi_1}. \tag{70}$$

Note that Lemma D.6 considers a vector $x$ uniformly distributed on the unit sphere, while in (70) $x$ is uniformly distributed on the sphere with radius $\sqrt{d}$. Thus, (63) can be re-written as

$$\mathbb{E}\left[ \exp\left( C \left| \sum_{\boldsymbol{i} \in [d]^r} a_{\boldsymbol{i}} \prod_{j=1}^r x_{i_j} \right|^{2/r} \right) \right] \leq 2. \tag{71}$$

By definition (60) of sub-exponential norm, we obtain that

$$\sup_{A\in\mathcal{A}}\left\|\left|\sum_{\boldsymbol{i}\in[d]^r} a_{\boldsymbol{i}}\prod_{j=1}^r x_{i_j}\right|^{2/r}\right\|_{\psi_1} = \frac{1}{C}, \tag{72}$$

which, combined with (70), leads to

$$\|\tilde{k}_x\|_{\psi_1} \leq C_1 d^{r/2-1}, \tag{73}$$

where $C_1$ is a numerical constant.

The third step is to bound the Euclidean norm of the rows of $\tilde{K}$. Recall that $\tilde{k}_x$ is obtained by keeping the elements of $x^{\otimes r}$ where the indices $i_1, i_2, \ldots, i_r$ are all different. As for the upper bound, we have that

$$\|\tilde{k}_x\|_2^2 \leq \left\|x^{\otimes r}\right\|_2^2 = d^r. \tag{74}$$

As for the lower bound, we have that

$$\|\tilde{k}_x\|_2^2 \geq \left\|x^{\otimes r}\right\|_2^2 - \left(d^r - \prod_{j=0}^{r-1}(d-j)\right)\left(\max_{i\in[d]}|x_i|\right)^{2r}, \tag{75}$$

since $x^{\otimes r}$ contains $d^r$ entries, $\tilde{k}_x$ contains $\prod_{j=0}^{r-1}(d-j)$ entries and each of these entries is at most $(\max_{i\in[d]}|x_i|)^r$. Note that $\prod_{j=0}^{r-1}(d-j)$ is a polynomial in $d$ of degree $r$ whose leading coefficient is 1. Thus,

$$\prod_{j=0}^{r-1}(d-j) \geq d^r - C_2 d^{r-1},$$

for some constant $C_2$ that depends only on $r$. Consequently,

$$\|\tilde{k}_x\|_2^2 \geq d^r - C_2 d^{r-1}\left(\max_{i\in[d]}|x_i|\right)^{2r}. \tag{76}$$

As $x$ is uniform on the sphere of radius $\sqrt{d}$, we can write

$$x = \sqrt{d}\frac{g}{\|g\|_2}, \tag{77}$$

where $g = (g_1, \ldots, g_d) \sim \mathcal{N}(0, I_d)$. Then,

$$\left(\max_{i\in[d]}|x_i|\right)^{2r} = \left(\frac{\sqrt{d}}{\|g\|_2}\right)^{2r}\left(\max_{i\in[d]}|g_i|\right)^{2r}. \tag{78}$$

Recall that the norm of a vector is a 1-Lipschitz function of the components of the vector. Thus,

$$\mathbb{P}\left(|\|g\|_2 - \mathbb{E}[\|g\|_2]| \geq t\right) \leq 2e^{-t^2/2}. \tag{79}$$

Furthermore,

$$\mathbb{E}[\|g\|_2] = \frac{\sqrt{2}\Gamma\left(\frac{d+1}{2}\right)}{\Gamma\left(\frac{d}{2}\right)}, \tag{80}$$

where $\Gamma$ denotes Euler's gamma function. By Gautschi's inequality, we have the following upper and lower bounds on $\mathbb{E}[\|g\|_2]$:

$$\sqrt{d-1} \leq \mathbb{E}[\|g\|_2] \leq \sqrt{d+1}. \tag{81}$$

As a result,

$$\mathbb{P}\left(\left|\left(\frac{\sqrt{d}}{\|g\|_2}\right)^{2r} - 1\right| > \frac{1}{2}\right) \leq 2e^{-C_3 d}, \tag{82}$$

for some constant $C_3 > 0$ depending on $r$ (but not on $d$). Consequently, with probability at least $1 - 2e^{-C_3 d}$, we have that

$$\left(\max_{i\in[d]}|x_i|\right)^{2r} \leq \frac{3}{2}\left(\max_{i\in[d]}|g_i|\right)^{2r}. \tag{83}$$

An application of Theorem 5.8 of [10] gives that, for any $t > 0$,

$$\mathbb{P}(\max_{i \in [d]} g_i - \mathbb{E}[\max_{i \in [d]} g_i] \geq t) \leq e^{-t^2/2}. \tag{84}$$

Furthermore, we have that, for any $\alpha > 0$,

$$e^{\alpha \mathbb{E}[\max_{i \in [d]} g_i]} \leq \mathbb{E}\left[e^{\alpha \max_{i \in [d]} g_i}\right] = \mathbb{E}\left[\max_{i \in [d]} e^{\alpha g_i}\right] \leq \sum_{i=1}^{d} \mathbb{E}\left[e^{\alpha g_i}\right] = d e^{\alpha^2/2}, \tag{85}$$

where the first passage follows from Jensen's inequality. By taking $\alpha = \sqrt{2 \log d}$, we obtain

$$\mathbb{E}[\max_{i \in [d]} g_i] \leq \sqrt{2 \log d}, \tag{86}$$

which, combined with (84), leads to

$$\mathbb{P}\left(\max_{i \in [d]} g_i \geq \left(\frac{d}{3C_2}\right)^{\frac{1}{2r}}\right) \leq 2 e^{-C_4 d^{1/r}}, \tag{87}$$

where $C_2$ is the constant in (76) and $C_4 > 0$ is a constant that depends only on $r$ (and not on $d$). Since the Gaussian distribution is symmetric, we also have that

$$\mathbb{P}\left(\max_{i \in [d]} |g_i| \geq \left(\frac{d}{3C_2}\right)^{\frac{1}{2r}}\right) \leq 4 e^{-C_4 d^{1/r}}. \tag{88}$$

By combining (76), (83) and (88), we obtain that, with probability at least $1 - 2e^{-C_5 d^{1/r}}$,

$$\|\tilde{k}_x\|_2^2 \geq \frac{d^r}{2}. \tag{89}$$

Hence, by doing a union bound on the rows of $\tilde{K}$, we have that, with probability at least $1 - 2N e^{-C_5 d^{1/r}}$,

$$\min_{i \in [N]} \|\tilde{K}_{i:}\|_2^2 \geq \frac{d^r}{2}, \tag{90}$$

$$\max_{i \in [N]} \|\tilde{K}_{i:}\|_2^2 \leq d^r, \tag{91}$$

where $\tilde{K}_{i:}$ denotes the $i$-th row of $\tilde{K}$.

The last step is to apply the results of Lemma D.7. Let $z = (z_1, \ldots, z_N) \in \mathbb{R}^N$ be such that $\|z\|_2 = 1$. Then,

$$\|\tilde{K}^T z\|_2^2 = \sum_{i=1}^{N} z_i^2 \|\tilde{K}_{i:}\|_2^2 + \sum_{i \neq j} \langle z_i \tilde{K}_{i:}, z_j \tilde{K}_{j:} \rangle, \tag{92}$$

which immediately implies that

$$\sigma_{\min}^2(\tilde{K}) \geq \min_{i \in [N]} \|\tilde{K}_{i:}\|_2^2 - B_N^2, \tag{93}$$

with

$$B_N = \sup_{z \,:\, \|z\|_2 = 1} \left| \sum_{i \neq j} \langle z_i \tilde{K}_{i:}, z_j \tilde{K}_{j:} \rangle \right|^{1/2}. \tag{94}$$

By applying Lemma D.7 and using the bounds (73) and (91), we have that

$$\mathbb{P}\left(B_N^2 \geq \max(C_6 d^{r-2} N, C_6 d^{r-1} \sqrt{N}, d^r/4)\right) \leq (1 + 3 \log N) e^{-11\sqrt{N}}, \tag{95}$$

for some constant $C_6$ depending on $r$. Recall that $N \leq c_1 d^2$ for a sufficiently small constant $c_1$ (which can depend on $r$). Thus, we have that

$$\mathbb{P}\left(B_N^2 \geq d^r/4\right) \leq (1 + 3 \log N) e^{-11\sqrt{N}}. \tag{96}$$

By combining (93), (90) and (96), we obtain that

$$\mathbb{P}\left(\sigma_{\min}^2(\tilde{K}) \geq d^r/4\right) \geq 1 - 2N e^{-C_5 d^{1/r}} - (1 + 3 \log N) e^{-11\sqrt{N}}, \tag{97}$$

which, together with (67), gives the desired result. □