[Reviews · NeurIPS 2020]

Review 1

Summary and Contributions: ------- Post author response -------------- Thank you for the response, you have addressed my questions and I am not changing the score. The difference between the NTK results and yours is interesting. Both results need the iterates to be close to the initialization but for different reasons. Importantly, as you mention, there are cases where the NTK approximation does not hold in your setting. I think it would be good to emphasize this in the revised version. Regarding the initialization, I understand the examples, but I am not sure if W_2 has to be exactly 0 in your first example, or it can be close to 0. For example, does the assumption hold with high probability if W_2 is initialized as IID Gaussian with zero mean and very small standard deviation? If it does, I think this strengthens the results, because this is "closer" to initializations used in practice and there are several theoretical results which assume this (e.g., papers which study the "rich regime" of training). --------------------------------------------------- In this work the authors prove that gradient descent trained on a pyramidal non-linear network, converges to a global minimum. The analysis is based on previous work ([26], [27]) that prove analytical expressions for the gradients and lower bounds on the gradient of the loss function. The main technical challenge is to show that the gradient is lower bounded by a positive constant and that the gradient is Lipschitz. Using these facts, the authors prove convergence to a global minimum using a well-known technique in non-convex optimization.

Strengths: I think that this work is interesting and provides a novel result in non-convex optimization of non-linear neural networks. The main novelty is that the result holds for a realistic pyramidal network topology. I found the analysis quite novel and I wonder how it relates to previous works that use NTK-type analyses (see questions below). I think this work should be accepted and I hope it will inspire future work on analyzing non-convex optimization of neural networks.

Weaknesses: It is not clear if the initialization in Section 3.1 is realistic or not (i.e., is it similar to something that is used in practice). What is a concrete example of an initialization that satisfies the conditions mentioned in this section? Some of the notations and statements are not clear: 1. Lemma 4.1 part 4 does not seem to be used in the proof of Theorem 3.2. Is it used? The convergence to the global minimum follows by the convergence to 0 loss and not using Lemma 4.1 part 4. Is this correct? 2. After line 92, does lambda_F^2 and lambda_F^3 correspond to lambda_F to the power of 2 and 3? If so, then why are both assumptions needed (both give lower bounds on lambda_F)? 3. Can gamma be equal to 0 in equation (2), such that we get the ReLU activation and not Leaky ReLU?

Correctness: The results seem to be correct.

Clarity: Mostly well written.

Relation to Prior Work: Yes

Reproducibility: Yes

Additional Feedback: I have a few questions: 1. How is the analysis in this work compared to NTK-type analyses? In the proof of Theorem 3.2 there is a bound on the distance of the weights from initialization. Do the weights stay close to initialization as in NTK? How does the NTK behave in your analysis? 2. What makes the assumption on the pyramidal network necessary for convergence? Why doesn't Lemma 4.1 hold for other networks? What is the intuition? 3. What is the challenge for extending these results to the cross-entropy loss? 4. What is the challenge for extending these results to non-smooth activations such as ReLU?


Review 2

Summary and Contributions: (Post-rebuttal) I have read the authors response. While I am generally satisfied with the response, I'd like to encourage the authors to work out the Xavier and minimum distance \phi > 0 case because this would allow for more straightforward comparisons to some existing results. =================== This paper studies the convergence of gradient descent (GD) on the squared error empirical risk of deep fully connected networks with some smooth activation functions (Assumption 2.2). Under architectural assumptions (Assumption 2.1) that the first layer is wider than the number data points N and that the remaining layers have a “pyramidal structure” (layers getting narrower towards the output layer), this paper provides a set of sufficient conditions on the initialization (Assumption 3.1) for GD to converge linearly to the global optimum (Theorem 3.2). In particular, this paper shows that using a rather unconventional initialization scheme that satisfies Assumption 3.1, width N in the first hidden layer suffices to assure global convergence of GD. For the popular Xavier initialization, the requirement on the width becomes \Omega(N^2 2^{O(L)} / \lambda_*^2), where L is the depth of the network and \lambda_* is the minimum eigenvalue of the “Gram matrix” of the output of the first hidden layer. For random sub-Gaussian data, the paper also provides high-probability lower bound on \lambda_* that is independent on the number of data points N and the dimension d (Theorem 3.3).

Strengths: This paper tackles a problem of great interest to the community and improves the existing results by reducing the width requirements for convergence and removing the restriction that all layers must be wide.

Weaknesses: The width improvement to N is achieved for a rather nonstandard initialization, while other results in the literature are established using the standard random initialization schemes (e.g. Xavier). Therefore, comparing the Xavier initialization result \Omega(N^2 2^{O(L)} / \lambda_*^2) with other existing results, the N^2 result is not entirely new to the community, except the difference in depth.

Correctness: I checked the proof in the main text and it looks correct to me.

Clarity: I think the paper is well-written in general, but it could benefit from providing some explanation on the conditions in Assumption 3.1. The conditions are very technical and difficult to parse; the paper does not provide any intuition as to what they mean and why they should hold.

Relation to Prior Work: The paper covers most of the known results, but misses some of the relevant results such as [1,2,3] (although [2,3] are not in the regression setting). Also, it would be better to note that for classification settings, there are results with polylog(N) width requirements (e.g. reference [22] in the paper), although under stronger assumptions on the data. Also, reference [17] in the paper achieves the convergence of perturbed gradient descent for networks of width smaller than N, albeit for shallow networks optimized over in only one layer; this work should be covered in more details in the paper. [1] Zhao Song, Xin Yang. Quadratic Suffices for Over-parametrization via Matrix Chernoff Bound https://arxiv.org/abs/1906.03593 [2] Amit Daniely. Neural Networks Learning and Memorization with (almost) no Over-Parameterization. http://arxiv.org/abs/1911.09873 [3] Amit Daniely. Memorizing Gaussians with no over-parameterizaion via gradient decent on neural networks. https://arxiv.org/abs/2003.12895

Reproducibility: Yes

Additional Feedback: I have a few comments and questions: - In Assumption 2.1, is n1 ≥ n2 also required, or is n1 < n2 allowed? - How do Assumption 3.1 and Theorem 3.2 look like in the case of L = 2? What is the corresponding initialization scheme (as in Section 3.1) when L = 2? - The initialization scheme in Section 3.1 requires that W_2 = 0 at initialization and \lambda_l ≥ c * (\bar \lambda_l / \lambda_l), which means that the W_l has large enough minimum singular value and be well-conditioned. Can you provide any intuition as to why this helps optimization? - Theorem C.4 (the Xavier case) requires that \sqrt{n_{l-1}} ≥ 1.01*(\sqrt{n_l}+t) for some t>0, which is actually stronger than Assumption 2.1! This should be explicitly mentioned in the main text. Also, Section 3.2 claims that the theorem holds for “any training data,” but what about some pathological cases such as duplicate data points (x_i, y_i), (x_j, y_j) satisfying x_i = x_j and y_i \neq y_j? - Can you obtain a similar convergence result with Xavier initialization and the assumption that \phi > 0 is the minimum L2 distance between any pair of training data points? If so, what is the width requirement in this case? - The sub-Gaussian norm || ||_{\psi_2} is used in Theorem 3.3 without definition; it should better be defined somewhere. - In Lemma 4.2, the notation “some scalars \bar \lambda_l” overloads with \bar \lambda_l in eq (3), adding confusion. Is the overload intended?


Review 3

Summary and Contributions: Post-rebuttal: I've read the author response and still recommend acceptance. The result is quite novel and of interest to the deep learning theory community. I highly recommend that, if accepted, the authors revise to provide a detailed comparison with the NTK literature and how their assumptions and results are related to the other state-of-the-art works. ********************** The authors consider a deep neural network with one wide layer followed by a sequence of thinner layers and demonstrate that gradient descent finds the global minimum. In contrast to previous works, they are able to show that for a certain initialization scheme, having width >= N = number of samples suffices, and show that N^2 suffices for the typical Xavier initialization scheme, without explicit NTK or mean field scaling.

Strengths: The paper makes a significant improvement on previous global convergence results in the neural network literature, providing the first result with overparameterization on the order of N, and for deep networks, the first of order < N^8, with essentially no assumptions on the training data. This work is addressed at the central question of the optimization of deep neural networks and seems to go beyond the NTK and mean field scaling approaches, which is significant. The proofs are presented in a clear manner and the paper is well structured.

Weaknesses: Some of these results are quite surprising and would benefit from a more detailed discussion on how the results should be reasonable, accompanied by relevant comparison with the assumptions and proof techniques of related literature. A few instances where this came up: (1) the results in Theorem 3.2 apply without any assumptions on the data, and thus could apply in the case that there are samples (x1, y1), (x2, y2) where x1 = x2 but y1 != y2, and more generally they would apply when there is no separability condition that is common in the works of e.g. Allen-Zhu [2] and Zou [46]. My understanding is that the initializations required for Thm 3.2 to go through would be impossible in the x1=x2, y1 != y2 setup, and similarly for Theorem C.4 lambda^*=0 in this case. But how does this work when the data is not separable and that in some sense x_i and x_j can be extremely close but have very different y_i, y_j? At a more general level, how is the initialization scheme related to the training samples for which you can guarantee convergence? (2) What happens when the data does not lie on the sphere of radius sqrt(d) but instead the sphere of radius 1? Do your results still hold then, or what's the issue? (3) Do you know how || theta(k) - theta(0) || behaves? I'm interested in if your methods are somehow still reliant upon the idea that the gradients of theta(k) are close to the gradients of theta(0) throughout the GD trajectory and so would still be somewhat in the `NTK regime' even if there is no explicit NTK scaling. If the authors are able to address these questions I think the paper will be quite improved and I will increase my score.

Correctness: I carefully checked the proofs outside of Sections B.1 and D.5 and they are correct.

Clarity: The paper is well-written.

Relation to Prior Work: The paper effectively surveys related work. In the weaknesses section above, I pointed out some places where a more detailed comparison with the techniques of other papers would be useful.

Reproducibility: Yes

Additional Feedback: l.171-172: Can you please cite which specific results in [26,27] derive the PL-type inequality in Lemma 4.1.3? I browsed through them and was unable to find these results, and indeed at first look it seems there are different assumptions on the activation functions in these papers. This result is essential to your proof so it is necessary to properly cite this. l.177: Can you give a sentence or two explaining why l=2 is not needed for point 4 of Lemma 4.1? It is not obvious to me. l.194-198: when citing (4) and (5), please also mention that this is a consequence of the choice of \alpha / the step size. Typos: l.10: the first sentence is a run-on sentence; need something to fix "[7], the optimization..." l.394: should be M_l^a, not M_L^a


Review 4

Summary and Contributions: ***post-rebuttal*** Thank you for your detailed answers. I would encourage you to include a discussion about the links and differences from the NTK literature into the main body of the final version of the paper, and trying to come up with a simple architecture where results of section 3.1 can be tested empirically for generalisation performance if time remains. *** This work studies convergence of gradient descent (GD) tasked with optimisation of neural network parameters with respect to squared error loss. The novelty of this paper is in establishing linear convergence rate of GD optimisation for neural networks with *pyramidal* architecture---i.e., first layer the widest, then decreasing width---as long as the number of neurons *in the first layer* is sufficiently large (the number of neurons in other layers may remain arbitrarily small). The authors further discuss how large the first layer has to be in order to satisfy the assumptions of their main result, and present an initialisation scheme that requires only order N (number of data points) neurons in the first layer. For Xavier initialisation, the authors show order N^2 neurons in the first layer suffices for linear convergence.

Strengths: - Pyramidal architectures are common (from VGG and ResNets, to EfficientNet) but not properly understood theoretically, making this a highly relevant contribution. - A substantial theoretical contribution where authors achieved similar or better dependence on N (number of data points) than `comparable' existing results (most existing results require all layers to be wide, hence the quotation marks).

Weaknesses: - There is no empirical evaluation. While fully theoretical papers are completely fine in general, this paper would benefit from showing whether the predictions they make hold in practice, and particularly for pyramidal networks of reasonable sizes (given that the main contribution of this paper is directed towards pyramidal networks). Relatedly, the result presented in section 3.1 is interesting but it would have been much more impactful if the authors could demonstrate it also leads to competitive *generalisation* error (the presented results only ensure the training loss shrinks to zero). - Assumption 2.2 excludes ReLU (differentiability), and if I understand correctly also sigmoid and tanh due to the assumption that the gradient of the activation is bounded away from zero.

Correctness: I did not check the proofs beyond high-level skimming.

Clarity: The paper was hard to follow at times. (However, this is partly a function of the high amount of presented technical content and the 8 page limit.)

Relation to Prior Work: Yes.

Reproducibility: Yes

Additional Feedback: - Can you please explain the relation between your work and the "lazy regime" typical for NTK? In particular, does the representation of the first layer evolve during training or tends to stay close to the initialisation? What about the subsequent layers (especially if they are far away from the order N^2 width)? - While I fully understand your results concern training loss and not generalisation error, I am wondering what implications do they have on the recent double descent literature which studies neural networks trained to zero training loss. Would you expect the double descent behaviour to also appear in pyramidal architectures?

[Author Response · NeurIPS 2020]

We thank the reviewers for the positive reviews and valuable feedback. First, we provide general comments addressing
remarks of multiple reviewers. Then, we reply to other remarks. We will update the manuscript accordingly.

**Connection to NTK** (Raised by Rev. 1, 3, 4). The idea of the proof is to combine the PL-inequality in Lemma 4.1
with the fact that the matrices $\{F_1, W_3, \ldots, W_L\}$ stay full rank during training. To show the latter, we prove that the
weights cannot move too far from initialization, see l. 192-195. Our non-convex optimization perspective allows us to
consider more general settings than existing NTK analyses. In fact, if the width of one of the layers is constant, then the
NTK is not well defined. On the contrary, our paper just requires the first layer to be overparameterized (i.e., all the
other layers can have constant widths).

**Training data and generalization** (Raised by Rev. 2, 3). As noted by Rev. 3, if $x_i = x_j$, then $\lambda_F = 0$. Thus,
Assumption 3.2 cannot hold unless $\Phi = 0$ (i.e., we initialize at a global minimum) or $X = 0$ (i.e., the GD iterates do
not move). In general, if the data points are not parallel and the activation function is analytic and not polynomial, then
$\lambda_* > 0$ (and thus, $\lambda_F > 0$), see [15]. Furthermore, if $x_i$ and $x_j$ are close, then $\lambda_F$ is small and, therefore, $\alpha_0$ is small.
Thus, GD requires more iterations to converge to a global optimum. This happens regardless of the value of $y_i$ and
$y_j$. Providing results for deep pyramidal networks that depend on the quality of the labels is an outstanding problem.
Solving it could also lead to generalization bounds, see e.g. "Fine-Grained Analysis of Optimization . . ." by Arora et al.

**Pyramidal network and spectrum of** $W_l$ (Raised by Rev. 1, 2). The pyramidal assumption is needed for Lemma 4.1.3
(l. 176). The key idea (see Lemma 4.3 in [27] for the proof) is that the norm of the gradient can be lower bounded by
the smallest singular value of $\prod_{p=3}^L A_p$ with $A_p = \Sigma_{p-1}(W_p \otimes \mathbb{I}_N) \in \mathbb{R}^{r_{p-1} \times r_p}$. Assuming that $r_2 \geq r_3 \geq \ldots \geq r_L$,
one can further lower bound this quantity by the product of the smallest singular values of the $A_p$'s. This is where our
assumption on the pyramidal topology comes from. Lemma 4.1 should be seen as providing a sufficient condition for a
PL-inequality, rather than suggesting that such a PL-inequality holds only for pyramidal networks. Intuitively, if $W_l$ has
large minimum singular value and is well-conditioned, then GD will keep it away from the zero-measure set of low-rank
matrices, in which case the loss satisfies the PL-inequality and has Lipschitz gradient, thus leading to convergence.

**Rev. 1.** *Initialization in Section 3.1:* Concretely, one can use Xavier's initialization for $W_1$, pick $W_2 = 0$ and
$[W_l]_{ij} \sim \mathcal{N}(0, (28c)^2/n_{l-1})$ under the extra assumption $\sqrt{n_{l-1}} \geq 2\sqrt{n_l}$ for sufficiently large $c$. This fulfils our
assumptions w.p. $\geq 1 - 2\sum_{l=3}^L e^{-n_{l-1}/32}$. Another option is to pick $W_l$ to be scaled identity matrices (or rectangular
matrices whose left block is a scaled identity). *Weakness 1:* The reviewer is right. Lemma 4.1.4 is not used explicitly,
it's only meant to add an interpretation to Lemma 4.1.3. *Weakness 2:* Yes, $\lambda_F^2$ and $\lambda_F^3$ correspond to the second and
third powers of $\lambda_F$. Our proof of Theorem 3.2 requires both lower bounds on $\lambda_F$. *Cross-entropy loss:* The challenge is
that this loss may not satisfy our PL inequality. Oftentimes a different analysis is required, which leads to stronger
assumptions on the data and weaker convergence guarantees, see e.g. [11, 22, 28, 36]. *ReLU:* Currently, ReLU does not
work because *(i)* its derivative is not Lipschitz, which is needed to prove (19), and *(ii)* it can have zero derivative, while
we need $\gamma > 0$ for the PL-inequality to hold. The second problem seems to us more fundamental, i.e. how to show a
PL-inequality for ReLU and ensure that it holds throughout the trajectory of GD.

**Rev. 2.** *Clarifications:* $n_1 < n_2$ is allowed; We will explicitly mention the condition $\sqrt{n_{l-1}} \geq 1.01(\sqrt{n_l} + t)$ for
the Xavier case; We will define the sub-gaussian norm. *Case $L = 2$:* Conditions (4)-(5) in Assumption 3.1 become
$\lambda_F^2 \geq 12 \|X\|_F \sqrt{2\Phi(\theta_0)} \max(\bar{\lambda}_1, \bar{\lambda}_2)$ and $\lambda_F^3 \geq 24 \|X\|_2 \|X\|_F \sqrt{2\Phi(\theta_0)}\bar{\lambda}_2$. To satisfy the first condition, one can
scale $W_1^0$ by a constant $c$, and set $W_2^0 = 0$. In fact, one can prove that $\lambda_F^2$ scales with $c^2$ for the class of activations
in (2), whereas the RHS scales with $c$. Thus, when $c$ is large enough, the first condition holds. Similarly, the second
condition also holds. As for Theorem 3.2, the expressions for $\alpha_0$, $Q_0$ and $Q_1$ simplify as the quantities involving $\lambda_l$ and
$\bar{\lambda}_l$ disappear for $l \in [3, L]$. *About $\phi > 0$:* We haven't worked out our bounds explicitly for this assumption, but this is
an interesting direction. One idea is to follow an approach similar to Appendix B of [29] to relate $\lambda_F$ to $\phi$. *Overloaded
notation:* Yes, it is intended. We want to apply the lemma for any upper bounds of $\bar{\lambda}_l$.

**Rev. 3.** *Weakness 2:* The same result holds if the data has unit norm and the weights of the first layer are scaled up
by a factor $\sqrt{d}$. The scaling of $x_i$ is chosen so that $\langle x_i, w_j \rangle \sim \mathcal{N}(0, 1)$, $w_j$ being the weight of the first layer. If this
is not the case, one would need to extend the Hermite analysis of Lemma D.3 in Appendix D.2. *Lines 171-172:* The
PL-inequality follows from the same argument of Lemma 4.3 in [27]. We apologize for the confusion. *Line 177:* Part 4
of Lemma 4.1 basically follows from part 3 by setting the gradient on the LHS to zero and, in the RHS of l. 176, $W_2$
only appears implicitly via the $\Sigma_l$'s matrices. *Lines 194-198:* Thanks! We will mention this. We will also fix the typos.

**Rev. 4.** *Double descent:* Thanks for an intriguing question. It is indeed possible (or even likely) that pyramidal networks
exhibit a nonmonotonic behavior in the test loss (double descent or even more complicated multi-scale phenomena as
in "The Neural Tangent Kernel in High Dimensions. . ." by Adlam and Pennington). One way forward is to study the
spectrum of the feature matrices at the different layers by extending the analysis of the paper mentioned above to the
pyramidal architecture. We regard this as a challenging (yet very interesting) open direction.

[Meta-Review · NeurIPS 2020]

This paper shows global convergence of gradient descent for deep neural networks that has wide first layer followed by pyramidal shape layers. It shows that an unconventional initialization with width N (data size) of the first layer suffices to show global convergence, which is much smaller than the required width for usual Xavier initialization. The presented result improves existing results greatly; the global convergence for width N is a great improvement from existing results. That is a valuable result. It is encouraged to add more detailed discussions about connection to existing NTK theories and possibilities of relaxing the assumptions maid in the analysis.